# Incomplete lytic cycle of a widespread *Bacteroides* bacteriophage leads to the formation of defective viral particles

Sol Vendrell-Fernández[1], Beatriz Beamud[2], Yasmina Abou Haydar[1],
Jorge Am de Sousa[3], Julien Burlaud-Gaillard[4], Etienne Kornobis[5,6], Bertrand Raynal[7],
Joelle Vinh[8], David Bikard [2]*, Jean-Marc Ghigo [1]*

1 Institut Pasteur, Université Paris-Cité, UMR CNRS 6047, Genetics of Biofilms Laboratory, Paris, France,
2 Institut Pasteur, Université Paris-Cité, UMR CNRS 3525, Synthetic Biology Laboratory, Paris, France,
3 Institut Pasteur, Université Paris-Cité, Microbial Evolutionary Genomics Laboratory, Paris, France,
4 Plateforme IBiSA des Microscopies, Université et CHRU de Tours, Tours, France, 5 Institut Pasteur,
Université Paris Cité, Plateforme Technologique Biomics, Paris, France, 6 Institut Pasteur, Université Paris
Cité, Bioinformatics and Biostatistics Hub, Paris, France, 7 Institut Pasteur, Université Paris-Cité, Molecular
Biophysics Platform, Paris, France, 8 ESPCI Paris, PSL University, UAR CNRS 2051, Biological Mass
Spectrometry and Proteomics, Paris, France

* david.bikard@pasteur.fr (DB); jmghigo@pasteur.fr (J-MG)

## Abstract

Advances in metagenomics have led to the identification of new intestinal temperate bacteriophages. However, their experimental characterization remains challenging due to a limited understanding of their lysogenic-lytic cycle and the common lack of plaque formation in vitro. In this study, we investigated the hankyphage, a widespread transposable phage of prominent *Bacteroides* symbionts. Hankyphages spontaneously produced virions in laboratory conditions even in the absence of inducer, but virions did not show any evidence of infectivity. To increase virion production and raise the chances of observing infection events, we identified a master repressor of the hankyphage lytic cycle, Rep-C$_{HP}$, whose silencing amplified hankyphage gene expression, and enhanced replicative transposition and virion production. However, attempts to infect or lysogenize new host cells with different capsular types remained unsuccessful. Transmission electron microscopy and capsid DNA sequencing revealed an abnormal virion morphology and incomplete DNA packaging of the hankyphage, suggesting that it cannot complete its assembly in laboratory conditions for reasons that are yet to be identified. Still, metavirome and phylogenetic analyses were suggestive of hankyphage horizontal transmission. We could also detect the activity of diversity-generating retroelements (DGRs) that mutagenize the hankyphage tail fiber, and likely contribute to its broad host range. This study sheds light on the life cycle of this abundant intestinal bacteriophage and highlights important gaps in our understanding of the factors required for the completion of its life cycle. Elucidating this puzzle will be critical to gain a better understanding of the hankyphage biology and ecological role.

provided the original author and source are credited.

**Data availability statement:** All individual quantitative observations (S1 to S18 Data) that underlie the data summarized in the figures and results of the manuscript are available at: https://figshare.com/s/3ff18c-c2f6cc1edab0ae. Supplementary tables are available at: https://figshare.com/s/e704aa15d732cbf7d1a4. RNA sequencing data are available at the EMBL functional genomics data collection (ArrayExpress) (accession: E-MTAB-14274). Genomic DNA data are available at ENA (accession: PRJEB85302). Hankyphage raw nanopore sequences are available at Arrayexpress (accession: E-MTAB-14724). Assembled and annotated hankyphages are available at the NCBI database (Bioproject accession: PRJNA1200653, Genbank accessions: PV107404-PV107409). Hankyphage capsid DNA sequencing data are available at ArrayExpress (accession: E-MTAB-14979).

**Funding:** J.-M.G was supported by Agence Nationale de la Recherche (https://anr.fr/), ANR 20CE15002201. J.-M.G., D.B., S.V.F. and B.B. received support from Agence Nationale de la Recherche, Laboratoire d'Excellence "Integrative Biology of Emerging Infectious Diseases", (https://labexibeid.fr/), ANR-10-LABX-62-IBEID. D.B was supported by the European Research Council (https://erc.europa.eu/homepage), 101044479, and E.K. was supported by Agence Nationale de la Recherche, (https://anr.fr/). Mass spectrometry equipment was subsidized by Conseil Régional d'Île-de-France (Sesame 2010 10022268, Sesame 2018 EX039194), ANR-10-INBS-09. None of the funders played any role in the study design, data collection, analysis, decision to publish nor the preparation of the manuscript.

**Competing interests:** The authors have declared that no competing interests exist.

**Abbreviations :** ACN, acetonitrile; ATCC, American Type Culture Collection; AUC, Analytical ultracentrifugation; CFU, colony-forming unit; dCas9, dead Cas9; ddPCR, digital droplet PCR; DGR, Diversity generating retro-element; GMM, Gut Microbiota MediumHP+, *B. thetaiotaomicron* hankyphage-containing strain; IGS, Intergenomic similarity; nt, non-targeting; RepC$_{HP}$, Hankyphage repressor C; RT, reverse transcriptase; TEM, transmission electron microscopy; TR, Template repeat; VR, Variable repeat.

## Introduction

The human gastrointestinal tract hosts a complex and dynamic community of bacteria, which interact with an equally rich population of bacteriophages [1]. These bacteriophages impose specific selection pressures and stimulate genetic exchanges, playing a key role in maintaining the stability and adaptability of the gut microbiome [2]. The current bloom in metagenomics provided vast amounts of bioinformatic data on the gut phageome, leading to the identification of new viral groups of both virulent and temperate bacteriophages [1]. However, the molecular characterization of these phages is just starting, and little is known about their biological roles, even for phages that infect abundant bacterial species [3].

*Bacteroides* are prominent members of human microbiomes providing multiple nutritional benefits and contributing to intestinal health and the development of the host's immune system [4]. *Bacteroides* are specifically targeted by various virulent bacteriophages [5–7] and are also hosts for crAssphages, the most abundant viruses within the human gut [8,9]. Several species of this genus are also lysogenized by the broad-range temperate hankyphage, originally identified in the *Phocaeicola dorei* HM719 strain (previously classified as *Bacteroides*), and then shown to be present in 13 different *Bacteroides* species [10]. These viruses possess DGRs, genetic elements that enable the hypermutation of variable regions (VRs) in associated genes [11]. Briefly, a non-coding RNA homologous to the VR is used as a template repeat (TR) by an error-prone reverse transcriptase (RT) that specifically makes mistakes at adenine positions [12,13]. The mutagenic cDNA then recombines with the variable region via a process that has not yet been elucidated [13]. DGRs were first described in *Bordetella* phage BPP-1 where they diversify the tail fiber gene, enabling host range modifications [14]. Additionally, hankyphages contain a putative transposase and may replicate via replicative transposition [10], contributing to the *Bacteroides* genetic diversity through various genomic rearrangements. Overall, little is still known about the regulation of the lysogenic-lytic decision, and the activity of their DGR elements in the *Bacteroides* hankyphages.

In this study, we investigated hankyphages found in the genome of the major gut symbiont *Bacteroides thetaiotaomicron*. We first characterized a collection of *B. thetaiotaomicron* hankyphages exhibiting DGR activity. We showed that hankyphage virions are spontaneously produced in absence of specific inducers. The lytic cycle of the hankyphage appears to be under the control of a CI-like transcriptional repressor that we named RepC$_{HP}$. Silencing *repc$_{hp}$* increased the expression of most hankyphage genes, and induced the phage's replicative transposition, DGR activity and virion production. However, the resulting virions did not lead to plaque formation nor detectable phage transfer events. Our results suggest that the lack of infectivity could be due to abnormal virion assembly and DNA packaging. Nevertheless, the presence of hankyphage across phylogenetically distant *Bacteroides* genomes, and evidence for their recent mobilization, suggest that at least some of these phages could form fully functional viral particles. Hence, our results provide insights into the lysogenic-lytic switch of a prominent yet understudied *Bacteroides* bacteriophage.

## Results

### Identification of *B. thetaiotaomicron* hankyphages with active DGR systems

To identify hankyphages integrated in *Bacteroides* genomes, we queried the genomic sequences of our previously described laboratory collection of 22 *Bacteroides* isolates (S1 Table [15],) with the reference hankyphage p00 from *P. dorei* HM179 [10]. Seven *B. thetaiotao*micron strains out of the 22 (32%) contained hankyphage-like prophages, hereafter called HP+ strains, which were verified by PCR using hankyphage-specific primers (S1A Fig). Additionally, these strains were subjected to long read sequencing to obtain the complete prophage regions (S3 Table). One

strain (jmh63) was discarded due to the presence of a fragmented hankyphage in the assembly (S1B Fig), resulting in six clinical lysogenic strains selected for further study. Their encoded hankyphages were generally conserved and mainly differed in the MuA-like transposase, putative tail fiber and DGR RT (Fig 1A). Additionally, all hankyphages grouped at the genus level (intergenomic similarity, IGS > 70%) with the reference hankyphage p00 (Fig 1A and S3 Table).

All hankyphage-like prophages encoded a predicted intact DGR system with the identification of identical TR sequences and diverse VRs (S3 Table) suggestive of DGR activity. To investigate whether hankyphage lysogenic strains grown in laboratory conditions underwent DGR mutagenesis, we analyzed sequencing data obtained from total DNA extracted from single overnight cultures. We then determined the nucleotide diversity ($\pi$) [18] in the VR region for the bacterial strains with at least 30 reads spanning the whole VR region (jmh42, jmh47, jmh51). We found that nucleotide diversity at positions targeted by the DGR, which corresponds to adenine residues in the TR region, was significantly higher than at non-targeted positions, with mutagenesis levels varying between strains (Fig 1B and 1C). This showed that the DGR diversifies the hankyphage tail fiber gene in our growth conditions.

## Hankyphage virions are spontaneously produced by *B. thetaiotaomicron* isolates

Evidence of hankyphage DGR activity among our *B. thetaiotaomicron* strains prompted us to test their potential virion production. For this, the supernatants of stationary cultures were filtered, and viral DNA was purified. We then used multiplex digital droplet PCR (ddPCR) to obtain absolute quantification of the number of encapsidated hankyphage DNA molecules using a probe and primers located on the DGR RT. The absence of remaining genomic DNA was further checked by probing for the bacterial housekeeping gene *rplB* (S2 Fig and S2 Table). As a positive control, we used the lysate of *P. dorei* HM719 (carrying the p00 hankyphage), and as negative controls, we used the lysates of a *B. thetaiotaomicron* strain lacking the hankyphage (HP-) (jmh44) as well as the lysate of a genetically amenable *B. thetaiotaomicron* strain (jmh43) in which we deleted the hankyphage (jmh43ΔHP). Using this method as a proxy for virion counting, we detected the presence of $10^6$–$10^8$ virions/ml in the filtered supernatant of all HP+ *B. thetaiotaomicron* strains (Fig 1D), indicating that hankyphage virions were produced spontaneously in overnight cultures.

Previous studies reported a lack of plaque formation by hankyphage p00 on *P. dorei* naïve strains [10]. Consistently, none of the lysates of our collection of HP+ *B. thetaiotaomicron* strains yielded plaques on any of our HP-free strains nor on the jmh43 strain cured from its hankyphage (jmh43ΔHP). To test if we could increase virion production and maximize the possibility of new infection events, we focused on the genetically amenable *B. thetaiotaomicron* jmh43 strain, which carries a hankyphage highly similar to the hankyphage p00 (IGS = 99%) (Fig 1A). We tested a panel of potential inducers, including presence of salts (BHISP), different concentrations of the SOS-response inducer mitomycin C, various antibiotics (carbadox, erythromycin, ciprofloxacin), oxidative stress ($H_2O_2$ or oxygen exposure), different media imitating gut conditions (Gut Microbiota Medium (GMM) [19] or MiPro [20]), relevant gut cues (bile salts, lactose), osmotic shock (NaCl), iron chelators (EDTA, 2,2′-dipyridyl (BIP)), different pH (pH5 or pH9) and different temperatures (30 or 42 °C). None of these conditions revealed a significant increase in hankyphage virion production (Fig 1E), and the virions obtained did not lead to any plaque formation on naïve *B. thetaiotaomicron* strains (jmh43 ΔHP and jmh44). Overall, our results indicated that hankyphage virions were spontaneously produced by all tested HP+ strains from our collection and that none of the tested potential inducers increased virion production in *B. thetaiotaomicron* jmh43.

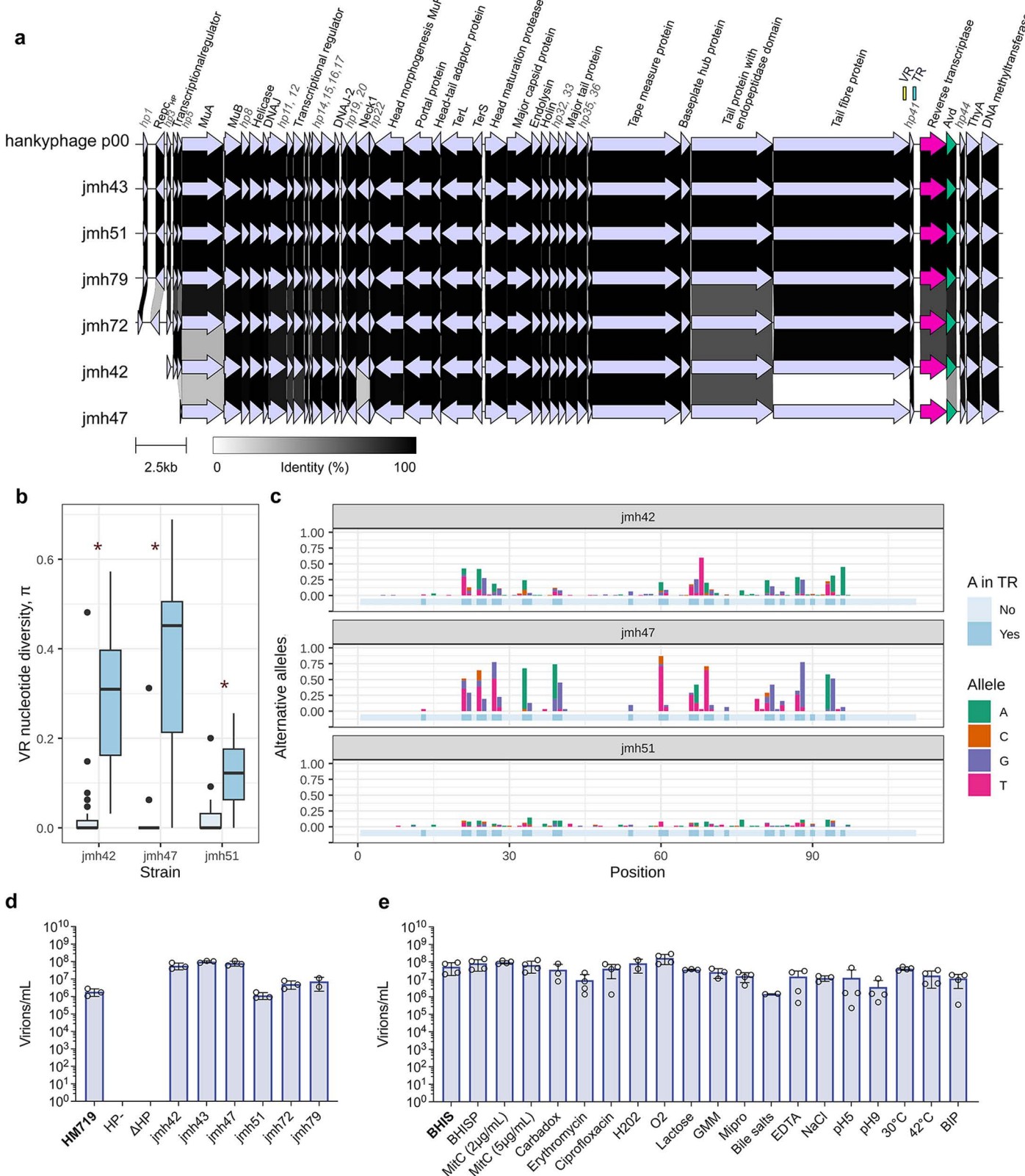

**Fig 1. Characterization of *Bacteroides thetaiotaomicron* hankyphage laboratory collection.** (**a**) Genome alignment of our laboratory *B. thetaiotaomicron* hankyphages as well as the newly annotated reference *Phocaelicola dorei HM719* hankyphage p00 using Clinker and Clustermap [16]. Gene annotations

for genes that had a predicted function are indicated in black whilst hypothetical genes are numbered and annotated in gray. Ortholog coding sequences (CDS) are linked through bars colored by the percentage of amino acid identity. The DGR system components are highlighted in different colors (VR: Variable repeat (yellow), TR: Template repeat (blue), reverse transcriptase (pink), and Avd: accessory variability determinant (green)). (**b**) Boxplot depicting the nucleotide diversity ($\pi$) of the hankyphage VR region for individual overnight cultures of HP+ *B. thetaiotaomicron* strains with >30× coverage (reads spanning the complete VR region). We considered a significant DGR activity (red asterisk) if the mean diversity of positions targeted by the DGR (A in the TR) was greater than the mean of non-targeted positions + 2 sd [17]. (**c**) Bar plots indicating the proportion of A, C, T, and G nucleotides at each VR position, colored if they differ from the reference VR of each strain. Positions targeted by the DGR system are highlighted in dark blue. (**d**) Bar plot indicating absolute virion counts in the lysate of non-induced HP+ strains obtained by ddPCR. The HM719 strain is indicated in bold as it corresponds to the strain with the reference hankyphage p00. Results from three independent biological replicates, points indicate individual values and bars indicate their average, error bars indicate standard deviance from the mean, calculated using Prism. (**e**) Bar plot indicating absolute virion counts in *B. thetaiotaomicron* jmh43 lysates after the addition of potential inducers obtained by ddPCR. Results from 3 to 4 independent biological replicates, points indicate individual values and bars indicate their average, error bars indicate standard deviance from the mean, calculated using Prism. BHIS, indicated in bold, corresponds to the basal induction levels in culture medium. After confirming their normality, statistics were performed using a one-way ANOVA (mixed effect analysis with Geisser-Greenhouse correlation) and individual comparisons to the BHIS control were performed using Dunnet's multiple comparison test. In the absence of indication, all comparisons showed non-significant differences. The individual quantitative values underlying this figure can be found in the S1 Data file also available at: https://figshare.com/s/3ff18cc2f6cc1edab0ae.

## The CI-like repressor RepC$_{HP}$ regulates hankyphage transcription

To increase virion production and raise the chances of observing infection events, we investigated if any genetic factors might trigger the hankyphage transition into the lytic cycle. For this, we looked for phage-encoded transcriptional regulators which could control the phage lysogeny. We noticed that one extremity of the hankyphage genome contained a small open reading frame, the location of which matched the position of the transposable Mu-bacteriophage repressor c (Repc) [21,22] (Fig 1B). This gene, therefore referred to as *repc$_{HP}$*, encoded a protein with a helix-turn-helix motif and shared a 37% amino acid identity to the λ phage CI repressor over the length of the full protein (Blastx). The Alphafold2 prediction of RepC$_{HP}$ as a dimer showed a strong structural alignment with the crystal structure of the N-terminal DNA-binding domain of the λ phage CI dimer (root mean square of atomic positions, RMSD value = 2.44), suggesting that RepC$_{HP}$ and the λ CI proteins could have similar DNA-binding properties (S3D Fig).

To test the involvement of *repc$_{HP}$* in the hankyphage virion regulation, we used the dead Cas9 (dCas9) technology [23–25] to inhibit *repc$_{HP}$* transcription in the HP + strain jmh43. We introduced a construct encoding an IPTG inducible *dCas9* gene and a guide targeting the *repc$_{hp}$* coding region in a permissive site located outside of the hankyphage prophage within the jmh43 chromosome. In this strain (jmh43::*dcas9-repc$_{HP}$*), *repc$_{HP}$* is expected to be silenced upon IPTG addition. As a control, we cloned a non-targeting (nt) control containing a guide that does not target any *B. thetaiotaomicron* genes [23] at the same location (jmh43::*dcas9-nt*). To test the potential repressor function of *repc$_{HP}$*, we extracted and sequenced total RNA from exponential growing cultures of jmh43::*dcas9-repc$_{HP}$* and jmh43::*dcas9-nt* before IPTG induction ($t = 0$ h) and then at 1, 2, 6 and 22 h post-induction (hpi). Consistently with the constitutive virion production that we detected in the WT *B. thetaiotaomicron* jmh43 strain, we observed hankyphage transcripts in both strains already at $t = 0$ hpi (S3A Fig). Differential RNAseq analysis between the two strains at each time point showed that upon IPTG induction, dCas9 effectively silenced the expression of *repc$_{HP}$*, and led to a significant increase in the expression of most hankyphage genes from 1 h post-induction onwards (Fig 2A). Altogether, these results suggest that the product of *repc$_{HP}$* is a repressor of the hankyphage gene expression that maintains lysogeny.

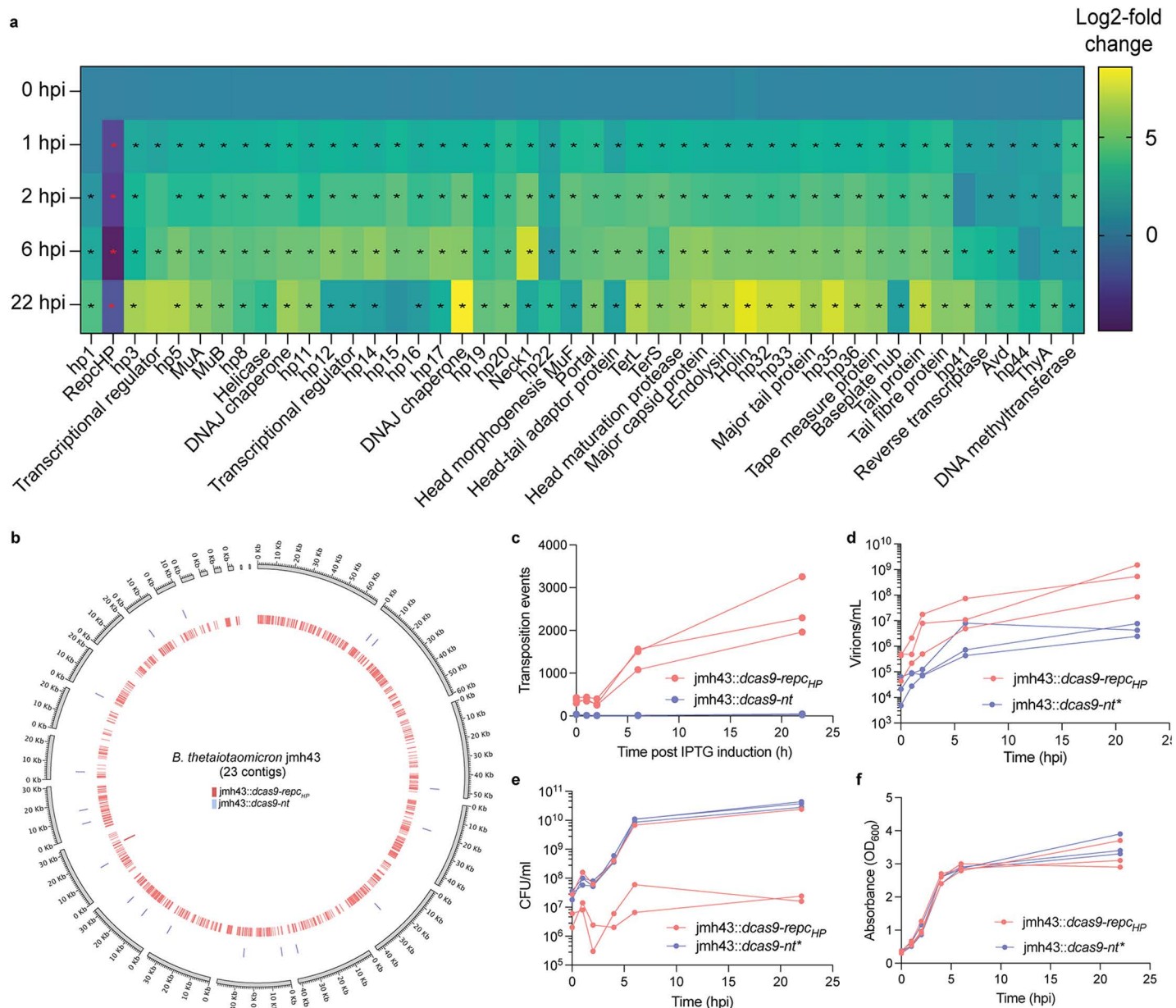

**Fig 2. The CI-like repressor RepC$_{HP}$ regulates hankyphage transcription, replication and virion production.** (a) Log2 fold change in gene expression along the hankyphage genome in the jmh43::*dcas9-repc*$_{HP}$ strain versus the jmh43::*dcas9-nt* control at different timepoints after IPTG-induced *repc*$_{HP}$ silencing. Stars indicate significant differential expression, either upregulation (black star) or down-regulation (red star) calculated by Deseq2 and with a cutoff of >1 log2 fold change and a p-adjusted value <0.05. (**b**) Mapping of transposition events in jmh43::*dcas9-repc*$_{HP}$ (red) and jmh43::*dcas9-nt* (blue) at 6 hpi (one biological replicate). (**c**) Quantification of transposition events of in jmh43::*dcas9-repc*$_{HP}$ and jmh43::*dcas9-nt* at different timepoints (three biological replicates). (**d**) Quantification of virion production by ddPCR, CFU/mL (**e**) and absorbance (**f**) of jmh43::*dcas9-repc*$_{HP}$ and jmh43::*dcas9-nt** across different time points (three biological replicates). The individual quantitative values underlying this figure can be found in the S2 Data file also available at: https://figshare.com/s/3ff18cc2f6cc1edab0ae.

## The RepC$_{HP}$ repressor regulates hankyphage replication and virion production

Phage master repressors typically trigger the expression of phage genes, initiating a cascade of events contributing to the phage lytic cycle, including replication of the phage genome,

virion assembly and exit from the host [22,26]. The hankyphage encodes a predicted MuA-like transposase and MuB-like ATPase [10] and it was hypothesized to replicate by transposition and insert randomly within the bacterial host genome, similarly to the Mu phage [10,27,28]. To test this, we investigated hankyphage replication following $repc_{HP}$ silencing. Concretely, from the same jmh43::$dcas9$-$repc_{HP}$ or jmh43::$dcas9$-$nt$ cultures from which we extracted RNA, we also extracted whole-cell genomic DNA and examined the flanking chromosomal regions present at each end of the hankyphage genome. This revealed the presence of insertion events throughout the genomes of both strains and at a much higher frequency in the jmh43::$dcas9$-$repc_{HP}$ strain compared to the jmh43::$dcas9$-$nt$ control strain (Fig 2B), at every time point. These insertions culminated at an average of 2,504 transposition events at 22 hpi, compared to an average of 40 events in the control (Fig 2C). This formally demonstrated that the hankyphage could replicate by replicative transposition and that $RepC_{HP}$ repressed phage replication.

To investigate if $repc_{HP}$ silencing also had an impact on virion production, we performed a similar experiment whereby we extracted and quantified hankyphage virions by ddPCR from jmh43::$dcas9$-$repc_{HP}$ and jmh43::$dcas9$-$nt$* (corrected strain, see material and methods section "*Correction of jmh43::dcas9-nt strain*") growing cultures after IPTG induction. This showed an overall increase in the number of virions detected in jmh43::$dcas9$-$repc_{HP}$ lysates compared to jmh43::$dcas9$-$nt$*, reaching an almost 1,000-fold increase at 22 hpi (Fig 2D). Silencing of $repc_{hp}$ was accompanied by a significant decrease in colony-forming unit (CFU) counts compared to the control (Fig 2E), but a lack of effect on optical density (Fig 2F). This suggests that induction of the hankyphage lytic cycle represents a strong burden for the cells without leading to substantial cell lysis. To determine whether DNA packaged into hankyphage virions showed signs of DGR diversification, we extracted and sequenced DNA from DNase-treated PEG precipitated capsids of jmh43 and jmh43::$dcas9$-$repc_{HP}$ lysates. We found DGR activity in both samples, with 7.93-fold (for jmh43::$dcas9$-$repc_{HP}$) and 3.32-fold (jmh43) more mutations at positions diversified by the DGR than non-targeted positions in the VR sequence (S4 Fig).

Overall, these results demonstrated that releasing $RepC_{HP}$ repression leads to the induction of the hankyphage lytic cycle, including replicative transposition and virion production with active DGR mutagenesis.

## Abnormal hankyphage virion assembly in jmh43 is partly recovered by $repc_{HP}$ silencing

To further examine the effects of $repc_{HP}$-silencing on hankyphage virions, we performed negative staining and transmission electron microscopy (TEM) on the lysates of jmh43 and jmh43::$dcas9$-$repc_{HP}$. TEM data presented in the initial description of the hankyphage p00 were suggestive of a tailed phage, consistent with the presence of predicted genes for neck and baseplate hub proteins [10]. Strikingly, we observed a total absence of tailed hankyphages in the lysate of jmh43 (Fig 3A). This lack of tails was also observed in the lysates of the reference *P. dorei HM719* strain (S6 Fig). Only in the jmh43-$dcas9$-$repc_{HP}$ lysate, three complete hankyphage particles could be observed from 10 different fields of view, as well as some loose tails. Both lysates also contained amorphous particles of different sizes ranging from 30 to 60 nm (S6 Fig) lacking the typical isometric appearance of mature capsids, which could be immature procapsids or vesicles (Fig 3A). Moreover, tailed hankyphages exhibited a greater capsid size than non-tailed particles (S7 Fig).

To confirm the nature of the different macromolecules present in the $repc_{HP}$-silenced lysate, we performed analytical ultracentrifugation (AUC) and analyzed the sedimentation profiles of the concentrated lysates of jmh43::$dcas9$-$repc_{HP}$ and the negative control jmh43ΔHP (Fig 3B).

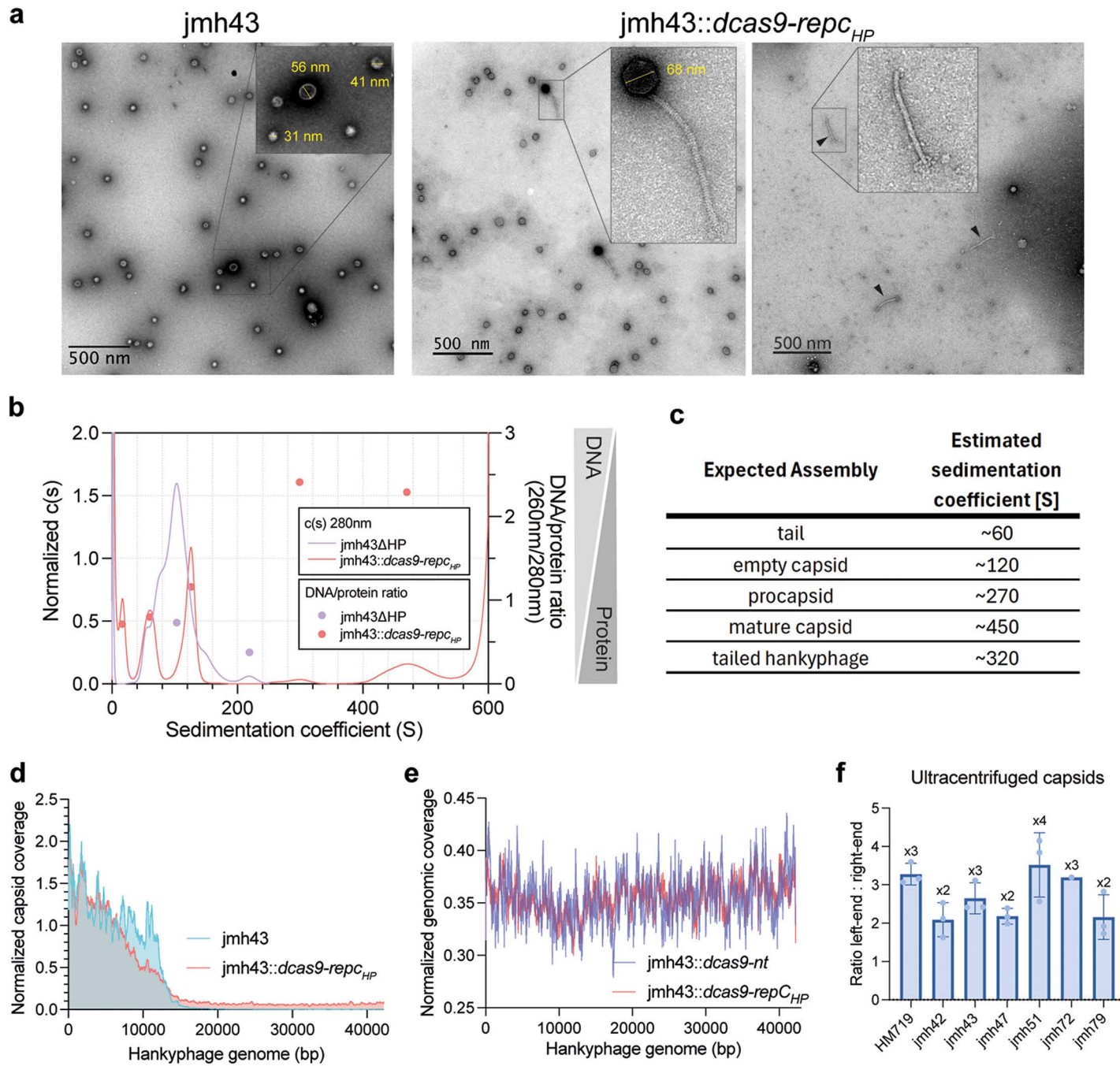

**Fig 3. Abnormal hankyphage virion morphology and incomplete DNA packaging.** (**a**) Negatively stained TEM micrographs of lysates of jmh43 and jmh43::*dcas9-repc*HP. Loose phage tails are indicated with black arrows. (**b**) Analytical ultracentrifugation sedimentation distribution of purified jmh43ΔHP and jmh43::*dcas9-repc*HP lysates. The 280 nm spectrum has been normalized by the sum of the values per sample and is plotted on the left *y*-axis. The 260/280 nm ratio, indicative of the DNA/protein ratio is plotted on the right *y*-axis and values are indicated circles color-coded by sample as indicated by the legend. The figure depicts one out of two representative biological replicates. (**c**) Estimated sedimentation coefficients for the different expected assemblies in the jmh43 and jmh43::*dcas9-repc*HP lysates. (**d**) Coverage map of DNA sequencing of PEG-precipitated and DNase-treated jmh43 and jmh43::*dcas9-repc*HP lysates. Reads mapped to the hankyphage genome and normalized by the sum of mapped reads to that region. (**e**) Coverage map of DNA sequencing of genomic DNA samples of the jmh43::*dcas9-repc*HP and jmh43::*dcas9-nt* strains at 22 hpi. Reads mapped to the hankyphage genome and normalized by the sum of mapped reads to that region. (**f**) Bar plot showing the ratio of hankyphage genome right-end and left-end copies quantified by ddPCR on ultracentrifuged capsids of HP+ strains. The individual quantitative values underlying this figure can be found in the S3 Data file also available at: https://figshare.com/s/3ff18cc2f6cc1edab0ae.

To decipher the type of assembly in each peak, the expected sedimentation values of different hankyphage assemblies were modeled using the dimensions observed in the negative stain micrographs (Fig 3C) (See methods section '*Analytical ultracentrifugation*'). For each peak, the 260/280 ratio was calculated to detect the presence or absence of DNA (Fig 3B). Different assemblies could be detected in the jmh43::*dcas9-repc*$_{HP}$ sample. One sedimented at ~62S with a high protein content (low 260/280 ratio), consistent with the expected sedimentation value of hankyphage tails. Another sedimented at 120S with a high protein content, consistent with empty capsids. Two broad peaks could be seen from 210S to 320S and then from 400S to 520S, both with a high DNA content, and which could correspond to procapsids or tail-less capsids, respectively (Fig 3B and 3C). The low abundance of fully assembled virions seen in our microscopy data makes it unlikely that we can detect them with AUC. These results agreed with the observation of detached tails and putative immature capsids in the jmh43::*dcas9-repc*$_{HP}$ negative staining images. The control jmh43ΔHP sample did not show any high DNA content particles; instead, it showed different assemblies with a low (<1) 260/280 nm ratio which were not found in the jmh43::*dcas9-repc*$_{HP}$.

The jmh43 strain encodes four additional predicted prophages, which can be potentially induced and confound results (S5 Table). Our RNAseq data detected expression for only some genes of these prophages and not of the full genome – in contrast to the hankyphage where all genes were expressed (S7 Fig). In line with this, mass spectrometry analysis of the *jmh43::dcas9-repC*$_{HP}$ lysate revealed the presence of most of the hankyphage proteins and an absence of proteins from the other four predicted prophages (S6 Table), suggesting that only hankyphage virions contribute to our observations. Additionally, we did not find an enrichment of coverage of other chromosomal regions beyond the hankyphage loci in the DNA extracted from the capsids for both jmh43 and jmh43::*dcas9-repc*$_{HP}$ strains.

Given the presence of some tailed virions in jmh43::*dcas9-repc*$_{HP}$ lysates, we decided to test their infectivity. However, plaque assays using jmh43::*dcas9-repc*$_{HP}$ lysates yielded no plaques on our HP-free strains nor jmh43ΔHP. Considering the role played by the *B. thetaiotaomicron* capsule for phage infection, we hypothesized that specific capsule types that might only be transiently expressed might allow hankyphage adsorption [5]. *B. thetaiotaomicron* contains 8 different capsule loci that are regulated in an on/off manner and generate extensive surface diversity [29]. We tested the infection potential of jmh43::*dcas9-repc*$_{HP}$ lysates as well as jmh43ΔHP lysates as a negative control, on a collection of capsule mutants of the hankyphage-free *B. thetaiotaomicron* VPI5482 strain, either expressing one of the eight single capsule types (cps1–8), none at all (△cps1–8), or heterogeneously expressing the eight (VPI548) [30]. We did not find any significant reduction in optical density for the non-capsulated mutant exposed to the jmh43::dcas9-*repc*$_{HP}$ lysate compared to the jmh43ΔHP lysate control (S8A Fig). Neither did we observe plaque formation on any of the mutants (S8B Fig).

Given the temperate nature of hankyphages, it could be possible that frequent and rapid lysogenization events impaired subsequent infections. We thus tested whether jmh43 and jmh43::*dcas9-repc*$_{HP}$ virions could lysogenize new strains. For this, we inserted an erythromycin-resistance encoding gene (*ermG*) with its own promoter on the *repc*$_{HP}$ end of the jmh43 hankyphage, downstream of *repc*$_{HP}$ in the WT strain (jmh43-HP-*Erm*) as well as the *repc*$_{HP}$ silenced strain (jmh43::*dcas9-repc*$_{HP}$-HP-*Erm*) (S8C Fig). We incubated the filtered supernatant of an overnight culture of strain jmh43-HP-*ermG*, containing *ermG*-tagged hankyphage, with HP-free strains (jmhΔ43ΔHP and jmh44) and the VPI5482 capsule mutants for 2 h followed by plating in the presence of erythromycin. PCR verification revealed that the few colonies obtained did not contain the *ermG* gene (S8D Fig), suggesting they probably were spontaneous HP-free erythromycin resistant mutants and indicating an absence of any new hankyphage lysogenization events. Additionally, to better mimic the gut's physiological conditions which might facilitate

phage infection, we performed the same transduction experiment using mouse fecal extracts, but this did not yield any transductants either.

Overall, our results show that the *B. thetaiotaomicron* jmh43 hankyphage spontaneously produces defective virions. However, despite the presence of complete virions after $repc_{HP}$-silencing, jmh43 hankyphages are non-infective in our laboratory conditions.

## Defects in hankyphage genome packaging in vitro

DNA sequencing of jmh43 and jmh43::*dcas9-repc*$_{HP}$ capsids revealed an enriched coverage in the entire hankyphage region, from which we could determine that this transposable phage packages 29–31 bp of the host chromosome on its 5′ end, and a variable fragment of 1–2.3 kb on its 3′ end, consistently with a headful packaging mechanism (S9 Fig).

Surprisingly, we observed that the hankyphage genome was abnormally encapsidated in the two samples, with the left, $repc_{HP}$-end (5′) of the hankyphage genome covered 40-fold times more than the right, methyltransferase (3′) end, with the coverage dropping gradually (Fig 3D). This contrasted with the uniform coverage obtained when sequencing total DNA extracted at different time points from the jmh43::*dcas9-repc*$_{HP}$ or jmh43 cultures (Fig 3E). Overall, our data shows that the jmh43 hankyphage DNA packaging is initiated at the left end of the phage but is randomly interrupted in the first 15 kb or ejected from the capsid, with only a small fraction of capsids being fully packaged. To investigate whether this phenomenon was specific to the hankyphage of strain jmh43 or a more general feature of hankyphages, we quantified the abundance of DNA at the left (MuA-like gene) and right end (DGR RT gene) of the hankyphage in virions present in the PEG-precipitated capsids of other HP+ *B. thetaiotaomicron* strains. In all the strains tested, the left end/right end ratio was even greater than the one detected in jmh43 (S10 Fig). To check that these differences were not due to virion damage during PEG concentration, we performed the same analysis after phage ultracentrifugation and found the same trend, albeit to a lesser extent (Fig 3F). Altogether, these results suggest that the hankyphages in all HP+ *B. thetaiotaomicron* strains of our collection have a packaging defect or are unstable in laboratory conditions.

## Genomic analysis of hankyphages suggests active mobility

Our results could suggest that at least some *B. thetaiotaomicron* and *P. dorei* hankyphages are no longer able to transfer horizontally due to structural defects. These defects could be the consequence of the burden potentially associated to the hankyphage spontaneous induction and transposition. However, deletion of the hankyphage generally did not significantly increase the growth of the host compared to the WT strains (S11 Fig), showing that the hankyphage does not pose a strong burden to the bacteria.

To gain a comprehensive understanding of the hankyphage distribution and potential mobility within *Bacteroides* and *Phocaeicola*, we compiled a dataset of 136 complete genomes from the NCBI, supplemented with the 25 strains from our laboratory collection (S4 Data). Out of these 161 strains, we found 26 (16%) encoding hankyphage-like prophages, with 20 of them grouping at the genus level (IGS > 70%) and 9 to the species level (IGS > 95%) with the reference hankyphage p00 [10]. Hankyphages closely related to this reference genome (at least 75% nucleotide identity, with most hankyphages having >95% nucleotide identity) were scattered across the phylogeny of *Bacteroides* and *Phocaeicola* (Fig 4). We then compared the prophage and bacterial phylogenies to test for potential evolutionary history correlation, and we observed that the phage phylogeny was not concordant with the phylogeny of the hosts and had multiple crossovers (Fig 4). Together with the presence of very similar hankyphage genomes in evolutionary distant bacterial hosts (S12 Fig), this suggests that there were recent

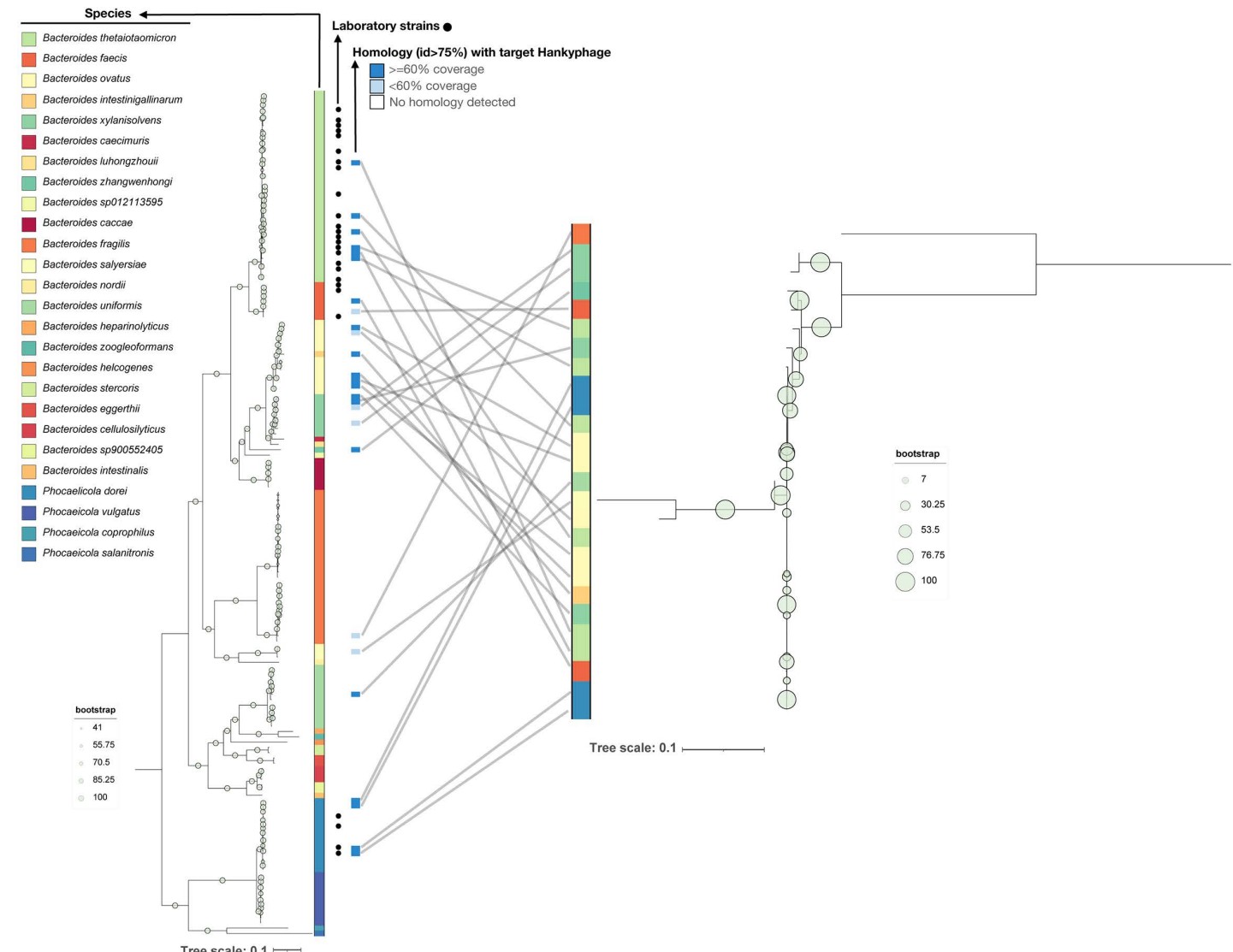

**Fig 4. Phylogenetic analysis of *Bacteroides* hankyphages.** Tanglegram showing *Bacteroides* (left) and hankyphage phylogenies (right) constructed from their respective core pangenomes [31], as well as crossover events representing the presence of Hankypages integrated in the genomes of the respective bacterial hosts. Hankyphage presence (identity above 75%, relative to the reference *Phocaelicola dorei* HM719 hankyphage p00) is indicated with a blue box, with darker blue boxes indicating a coverage of at least 60%, and lighter blue boxes indicating a coverage of less than 60% of the reference hankyphage genome. Tree files for *Bacteroides* and hankyphages and their respective IDs can be found in supplementary S17, S18 and S4 Data, respectively, available at: https://figshare.com/s/3ff18cc2f6cc1edab0ae.

events of hankyphage horizontal transmission. Moreover, analysis of the coverage pattern of hankyphages in metaviromes [10] did not reveal a drop in coverage toward the 3′ end of the hankyphage genome (S13 Fig). This suggested that the defect in capsid DNA packaging that we observe in our laboratory conditions does not occur in vivo. Overall, these results support the hypothesis that hankyphages are active and can transfer horizontally.

## Discussion

Hankyphages are widespread temperate phages found across different genera of the *Bacteroidaceae* family [6,10,32]. Despite their prevalence, their activity and factors regulating their potential lysogenic-lytic transition are poorly understood. In this study, we investigated the

hankyphage biology in a collection of *B. thetaiotaomicron* isolates, one of the most abundant *Bacteroides* species of the human gut microbiota.

Our study demonstrated hankyphage activity by detecting spontaneous virion induction in all tested HP+ *B. thetaiotaomicron* strains. First described by Lwoff in 1953, spontaneous induction by lysogens has now been observed in a variety of lysogenized bacteria [33]. For instance, a recent study that systematically detected temperate bacteriophage virion production from gut microbiota samples showed that a portion of Bacteroidota and Pseudomonadota phages produced virions, with titers only slightly enhanced after the addition of inducers [32]. Continuous production of induced phage particles was suggested to maintain the prophage ability to replicate and avoid degradation and loss [33]. Alternatively, chronic virion release was shown to enhance the fitness of the bacterial population, as it is the case for Shiga-toxin-encoding phages that prime epithelial cells for infection by STEC *Escherichia coli* [34].

Previous studies reported the induction or increase of hankyphage virion production by carbadox or $H_2O_2$ in *P. dorei* [10,32]. We could not confirm these results in *B. thetaiotaomicron,* nor identify any other conditions inducing hankyphage production. However, we showed that hankyphage virion production in *B. thetaiotaomicron* is regulated by $repc_{HP}$, a transcriptional repressor displaying homology with the λ CI repressor. Silencing of $repc_{HP}$ led to increased expression of most hankyphage genes, as well as replicative transposition, virion production and hankyphage tail formation. This is similar to the Mu-phage repressor c, where point mutations in this master repressor trigger the regulatory lytic cycle cascade [35]. Similarly to phage Mu, the hankyphage employs a headful packaging mechanism that leads to the packaging of about 2 kb of bacterial DNA on the right end of the phage, but of 29–31 bp on the left end, compared to 50–150 bp for Mu [36]. We could further observe that the DGR system carried by the hankyphage actively diversifies the variable region of the tail fiber gene.

Despite observing an increase of the hankyphage replicative and lytic cycle upon $repc_{HP}$ silencing, we did not obtain any evidence for secondary infection nor lysogenization. Although the absence of plaque formation has also been described in other abundant *Bacteroides*-infecting phages such as ΦcrAss002 [37], examination of the hankyphage virion morphology and DNA content revealed two possible defects. First, heads were smaller than expected and not attached to tails. This lack of tails was also observed in the lysates of the reference *P. dorei HM719* strain. Second, the hankyphage genome was only partly packaged. These observations suggest an incomplete capsid maturation process resulting in either partial DNA packaging or DNA ejection and its subsequent degradation upon DNase treatment. Incomplete capsid maturation might also prevent tail-capsid attachment, which could explain the high frequency of detached capsids and tails in the $repc_{HP}$ silenced strain. The proPhi1 phage and several other *Lactococcus lactis* phages were also shown to spontaneously release tailless virions, but the addition of mitomycin C resulted in increased virion production and the restoration of tail formation [38,39]. Perhaps unidentified bacterial-specific triggers or gut-specific signals are necessary for the complete assembly of hankyphage mature particles. Notably, some intestinal prophages remain uninduced in stool samples whilst being active within their hosts [40] and the transduction of certain *Bacteroides* phages has only been observed in the mouse gut [7].

The low frequency of complete hankyphage packaging could also be due to competition with other mobile genetic elements or anti-phage defense systems within the host [41–43]. Importantly, though the jmh43 strain contains 4 other prophage-like regions (S5 Table), sequencing the DNA content of the capsids and mass-spectrometry data did not detect other predicted prophages apart from the hankyphage.

Six anti-phage defense systems (S7 Table) could also be detected in the genome of jmh43 [44] and were transcriptomically active according to our RNAseq data (S14 Fig). Predicted

defense systems included a restriction-modification system (BREX), abortive infection systems such as PARIS or AbiE, but also defense systems with unknown mechanisms as PD-Lambda, PsyrTA, or Mokosh [45–48]. Additionally, we identified a Bil-like defense system, a recently described system shown to hinder phage assembly through the ubiquitination of target phage tails [49]. However, the deletion of this system did not affect the hankyphage assembly (S15 Fig). We cannot exclude that some of the other defense systems detected or that unknown anti-phage systems might be responsible for the defects in hankyphage assembly. A recent study demonstrated how defense systems carried by prophages can block the lytic cycle and release of competing prophages present in the bacteria [50]. A future direction will be to delete defense systems and prophages in the *B. thetaiotaomicron* host to test this hypothesis.

Moreover, beyond the putative prophages and defense systems, the expression of several hundred genes was detected to be significantly altered in the jmh43::*dcas9-repc*$_{HP}$ strain compared to the control strain at 6 hpi and 22 hpi (S2 Data). Even though the high number of hypothetical genes did not allow for an informative KEGG analysis, these results highlight the possibility of other cellular mechanisms interfering with hankyphage assembly or being affected by hankyphage induction.

Another explanation for the inability of the hankyphage to fully assemble could be that the hankyphage is simply defective. However, our phylogenetic analyses showed that hankyphages are present in various and distant parts of the *Bacteroides* tree, and that hankyphages likely underwent horizontal transfers. Although such transfer has not been demonstrated in vitro, a recent study that followed the microbiota composition of mothers and newborns before and after birth observed mobilization of the hankyphage across different *Bacteroides* species [51]. Finally, given the transposition ability of the hankyphage, it is possible that this phage could be mobilized horizontally by inserting into other mobile genetic elements such as plasmids and/or integrative and conjugative elements.

In summary, this study identified a key regulator of the hankyphage lytic cycle and several structural abnormalities that might hinder secondary infections in *B. thetaiotaomicron* strains. Future work will determine the mechanism driving the constitutive production of immature capsids, investigate whether they play any biological role in the hankyphage-*Bacteroides* interactions in the gut, and look for possible environmental cues that could drive the assembly of full hankyphage particles in the gut environment.

## Materials and methods

### Bacterial strains and growth conditions

*Bacteroides* strains were grown in brain-heart infusion medium supplemented (BHIS) with cysteine 0.1%, hemin 5 μg/mL and sodium bicarbonate 0.2% (NaCOH$_3$). Cultures were incubated statically at 37 °C in a C400M Ruskinn anaerobic chamber (95% N$_2$, 5% CO$_2$) or in anaerobic bags (GenBag anaero, Biomerieux). *E. coli* strains were grown in Luria-Bertani (LB) broth (Corning) supplemented with or without ampicilin 100 μg/mL and incubated at 37 °C aerobically with 180 rpm shaking.

### Construction of *B. thetaiotaomicron* marker-less deletion mutants

Deletion mutants of *B. thetaiotaomicron* were performed by allelic exchange using the pLGB13 vector [52]. Briefly, 500 base pairs upstream and downstream regions of the target sequence were cloned into the pLGB13 vector by Gibson assembly and transformed into *E. coli* S17 λpir. This strain was then used to deliver the vector to *B. thetaiotaomicron* by conjugation. Matings were carried out by mixing exponentially grown cultures

of the donor and the recipient strains at a 2:1 ratio and placing the mixture on BHIS agar plates at 37 °C under aerobic conditions overnight. The next morning, bacteria were plated on selective BHIS agar supplemented with erythromycin (15 µg/mL) for selection of *B. thetaiotaomicron* transconjugants that underwent the first recombination event and gentamicin (200 µg/mL) to ensure the exclusion of *E. coli* growth. Erythromycin resistant colonies of *B. thetaiotaomicron* were then grown overnight without antibiotics to promote plasmid loss and subsequently subjected to a second round of selection by plating on BHIS agar plates supplemented with anhydrotetracycline (for selection of double recombined colonies). The resulting deletion mutants were confirmed by PCR using external primers and by sanger or nanopore amplicon sequencing, or whole-genome illumina sequencing.

### dCas9-mediated silencing

An easily modifiable vector based on the pMM704 plasmid [23] was constructed by Gibson assembly where all BsaI sites were removed and an mCherry gene flanked by bsaI sites was introduced at the position of the guide insertion. Oligonucleotides were then designed to allow cloning of the desired guide between two BsaI sites on the vector. Oligonucleotides were phosphorylated and annealed to reprogram the guide carried by the vector using the Golden Gate assembly method. The mixture was then dialyzed and introduced into *E. coli* S17 λpir by electrophoration. The clones were confirmed by PCR and nanopore sequencing and if correct, were conjugated into *B. thetaiotaomicron* as described above. Bacteria were then selected on selective BHIS agar supplemented with erythromycin (15 µg/mL) and gentamicin (200 µg/mL) to ensure the exclusion of *E. coli* growth and the resulting mutants were confirmed by PCR using external primers and sanger or nanopore amplicon sequencing, or whole-genome illumina sequencing.

### Correction of jmh43::*dcas9-nt* strain

Differential expression analysis on the RNAseq data revealed a differential over-expression of the *dcas9* gene in jmh43::*dcas9-repc*$_{HP}$ compared to the jmh43::*dcas9-nt* strain (S3C Fig), which led us to find a 6-bp deletion in one of the lac operators of the IPTG-inducible promoter controlling the expression of dCas9 in jmh43::*dcas9-nt*. This strain was re-constructed to correct the promoter sequence (jmh43::*dcas9-nt**) (S4 Table) and further experiments were performed with both versions of the strain, yielding similar results.

### Hankyphage induction

Overnight cultures were diluted to an absorbance of $OD_{600} = 0.05$ and grown anaerobically in 5 mL cell culture tubes at 37 °C until exponential phase. For time point experiments, cultures were grown in 20 mL volume in 50 mL falcon tubes. Inducers were added to the following concentrations: BHISP (standing for BHIS-Phage media, contained the following salts: $CaCl_2$ 1 mM and $MgSO_4$ 10 mM), mitomycin C 2 µg/mL or 5 µg/mL, carbadox 8 µg/mL, erythromycin 5 µg/mL, ciprofloxacin 0.1 µg/mL, bile salts 0.5%, NaCl 0.5 M, EDTA 0.2 mM, bip (2,2′-dipyridyl) 0.2 mM, hydrogen peroxide ($H_2O_2$) 0.5 mM, lactose 0.4%, 42, 30 °C. For the cultures induced in GMM or MiPro medium, overnight cultures were adjusted to an OD600 = 0.3 and then washed and resuspended in either medium supplemented with 0.4% glucose. The cultures were then incubated anaerobically at 37 °C for 18 h. The cultures were then centrifuged at 7142*g* for 15 min at 4 °C, the supernatant was then filtered using 0.45 µm filters and the lysate stored at 4 °C.

## Lysis curves

Overnight cultures were diluted to an absorbance of $OD_{600} = 0.05$ and grown anaerobically in 5 mL cell culture tubes at 37 °C for 2 h until exponential phase. Lysates of different strains were mixed in a 1:1 ratio in a total volume of 200 µl in 96-well Greiner flat-bottom plates with the exponentially-grown bacterium inside the anaerobic station, and then sealed with an adhesive sealing sheet (Thermo Scientific, AB0558). The plates were then incubated in a TECAN Infinite M200 Pro spectrophotometer for 24 h at 37 °C. Absorbance at $OD_{600}$ was measured every 15 or 30 min, after a 200-second orbital shaking with a 2 mm amplitude.

## Transductions

Overnight cultures were diluted to an $OD600 = 0.05$ and grown anaerobically in 5 mL cell culture tubes at 37 °C for 2 h until exponential phase. One mL of cells were then pelleted and were either resuspended with 200 µl of the phage lysate or in a 1:1 ratio mixture of the phage lysate and mouse faecal filtrate. These were then incubated anaerobically and statically at 37 °C for 2 h. The mixture was then plated on erythromycin-supplemented BHIS plates and grown for 48 °C at 37 °C anaerobically. The colonies obtained were re-streaked on erythromycin-supplemented BHIS and single colonies were tested by PCR for the presence of the hankyphage and the *ermG* gene.

## CFU counts

Cultures were diluted serially in 10-fold or 100-fold dilutions in BHIS and 5 µl drops were placed on BHIS or BHIS-erythromycin solid media and grown for 48 h at 37 °C anaerobically. Colonies were counted from the most serially diluted spot containing >3 and <30 single colonies. The CFU was calculated using the following formula: CFU/mL = ((Number of colonies * total dilution factor)/ volume of culture plated).

## Growth curves

In Greiner flat-bottom 96-well plates, overnight cultures were diluted to a final absorbance of 0.1 ($OD_{600}$) within 150 µL BHIS in anaerobic conditions and a plastic adhesive film (adhesive sealing sheet; Thermo Scientific, AB0558) was used to seal the plate. The plates were then incubated in a TECAN Infinite M200 Pro spectrophotometer for 24 h at 37 °C. $OD_{600}$ was measured every 15 min, after a 200-s and 2 mm amplitude orbital shaking.

## Virion quantification by digital PCR (ddPCR)

Phage lysates were DNase treated to remove remains of genomic DNA by adding 2 µL of TURBO DNase (Thermo Fisher, REF #AM2238) and 10 µL of 10× TURBO Dnase Buffer to 88 µL of lysate, incubating at 37 °C for 1 h and then heat-inactivating the enzyme and opening the phage capsids by a 10-min 99 °C treatment. ddPCRs were performed as triplex reactions. For quantification of the virion production of HP+ strains, as well as the quantification of hankyphage right-end and left-end copies, we used a probe and primers specifically targeting a conserved region of the hankyphage *muA* gene and another targeting the *rt* gene, as well as the *rplB*-targeting genomic control (S2 Table). Reactions on jmh43 lysates obtained from different inducers as well as on jmh43::*dcas9-repc*$_{HP}$ and jmh43::*dcas9-nt* were performed with a probe and primers specific to the hankyphage *rt gene*, a second probe specific for the housekeeping gene *rplB*, and a third probe specific for the 3′ flanking region of hankyphage and the jmh43 chromosome (S2 Table).

Reactions were prepared either using the Naica PCR mix (R10056) or the Perfecta Multiplex qPCR ToughMix (Quanta Biosciences, Gaithersburg, MD, USA) and placed on either Sapphire or Ruby chips (Stilla technologies). Digital PCR was conducted on a Naica Geode using the following steps: droplet partition (40 °C, atmospheric pressure AP to + 950 mbar, 12 min), initial denaturation (95 °C, + 950 mbar, for 2 min), followed by 45 cycles at (95 °C for 10 s and 58 °C for 30 s), droplet release (down 25 °C, down to AP, 33 min). Image acquisition was performed using the Naica Prism6 reader. Images were then analyzed using the Crystal Reader (total droplet enumeration and droplet quality control) and the Crystal Miner software (extracted fluorescence values for each droplet) (Stilla Technologies). Values were then transformed from copies/μl to copies/mL.

## Identification and annotation of hankyphages

All complete *Bacteroides and Phocaeicola* genomes from the NCBI were obtained with ncbi-genome-download (accession: 08/08/23) [53]. Additionally, we included 22 *Bacteroides* isolates available in our laboratory [15] and four *P. dorei* strains obtained from the Biodenfense and Emerging Infections (BEI) via the American Type Culture Collection (ATCC), resulting in a total of 161 genome sequences (S4 Data). The classify workflow from GTDB-Tk v2.1.1 [54] was used to correct the taxonomy classification of the bacterial genomes. geNomad [55] was used to detect prophages which were then queried against the hankyphage reference genome (BK010646, 42,831 bp) using blastn [56] (-evalue 1e-5 -max_hsps 1). Candidate hankyphage-like prophages were extracted from hits with a minimum alignment length of 10% with the reference phage with seqtk (https://github.com/lh3/seqtk). A blastn search independent of geNomad identification revealed some additional HP+ clinical strains. In these strains, the phage was highly fragmented in the assembly, which prevented its detection by geNomad in the initial analysis. HP + laboratory strains were subjected to nanopore sequencing to improve the quality of the genome assembly (ArrayExpress accession number: E-MTAB-14724). Hybrid assemblies of Illumina and nanopore reads were obtained with Unicycler v. 0.4.8 [57,58]. Hankyphage-like prophages were extracted as above and further annotated with pharokka [58] (-p prodigal) and phold, an annotation pipeline that uses protein structures to annotate phage genes [59,60]. Genes with low confidence were manually curated using Foldseek [61]. Annotated hankyphages can be found at the NCBI database, accession: PRJNA1200653. Extracted hankyphage-like prophages were further compared with VIRIDIC which calculates IGS to classify phages at the genus and species level [62]. Clinker was used to obtain orthologous genes and comparison between hankyphage-like genomes [16]. Defense systems were identified with DefenseFinder [44,63,64].

## Alphafold2 modeling of RepC$_{HP}$

The complete amino acid sequence of RepC$_{HP}$ from the jmh43 hankyphage was submitted to the Alphafold2 collab server as a dimer with the following parameters: num_relax = 0, template_mode=none. The model with highest confidence was aligned to the N-terminus fragment of a dimeric crystal structure of the λ CI repressor (PDB3kz3) [65] using the PyMOL function "align".

## RNA isolation and sequencing

Overnight cultures were mixed with RNAprotect (Qiagen) in the anaerobic chamber to prevent RNA degradation and bacteria were lysed using QIAGEN Proteinase K and TE buffer containing lysozyme. Total RNA was extracted using the Direct Zol kit (Zymo Cat. R2050) according to the manufacturer's instructions and treated with Dnase I from the same kit. RNA

concentration, quality, and integrity from three independent replicates were checked using the 4195 Tapestation system (Agilent).

## RNAseq analysis

Sequencing was performed by the Biomics platform at the Institut Pasteur. Ribosomal RNA depletion was performed using the Bacteria RiboZero Plus kit (Illumina) with a *Bacteroides thetaiotaomicron* custom ribodepletion probe. From rRNA-depleted RNA, directional libraries were prepared using the Illumina Stranded total RNA Prep ligation preparation kit following the manufacturer's instructions (Illumina). Libraries were checked for quality on Fragment Analyzer (Agilent). Quantification was performed with the fluorescent-based quantitation Qubit dsDNA HS Assay Kit (Thermo Fisher Scientific). 750pM of pooled library were sequenced in pair-end 100 cycles on a NextSeq2000 Illumina sequencer. The RNA-seq analysis was performed with Sequana 0.14.1 [66]. We used the RNA-seq pipeline 0.15.2 (https://github.com/sequana/sequana_rnaseq) built on top of Snakemake 7.8.5 [67]. Briefly, reads were trimmed from adapters using Fastp 0.20.1 [68] then mapped to the jmh43::*dcas9-repc*$_{HP}$ genome (extracted at time point 0 hpi) using bowtie2 v.2.4.2 [69]. The reference jmh43::*dcas9-repc*$_{HP}$-t0hpi genome (accession: ERR14225767) was annotated using Prokka [70] and the hankyphage hypothetical genes were attributed specific names based on their coordinates. FeatureCounts 2.0.1 [71] was used to produce the count matrix, assigning reads to features using corresponding genome annotation with strand-specificity information. Quality control statistics were summarized using MultiQC 1.11 [72]. Statistical analysis on the count matrix was performed to identify differentially regulated genes. Clustering of transcriptomic profiles were assessed using a Principal Component Analysis. Differential expression testing was conducted using DESeq2 library 1.30.0 [73] scripts indicating the significance (Benjamini-Hochberg adjusted $p$-values, false discovery rate FDR < 0.05) and the effect size (fold-change) for each comparison. A cutoff of a minimum of 1 log2 fold change and a $p$-adjusted value <0.05 was set. Differential expression results on the hankyphage genes alone or all jmh43 genes are presented in supplementary S2 Data and S6 Data respectively. Raw reads can be accessed at the EMBL functional genomics data collection (ArrayExpress) (accession: E-MTAB-14274).

## Genomic DNA sequencing

Whole-cell genomic DNA was extracted from 2 mL of pelleted cultures using the Qiagen Blood and Tissue Kit (REF #69504) following the instructions for Gram negative bacteria. Samples were then RNAse-treated and resuspended in 50 µl of DNase and RNase-free water. The samples were then sequenced at the Institut Pasteur Biomics platform using a Nextseq2000 Illumina sequencer with a P1 flowcell. Libraries were prepared using the Illumina Truseq DNA PCR-Free kit. Sequencing was performed as paired-end in 300 cycles.

## Quantification and mapping of transposition events

Raw reads from genomic DNA (available at ENA—accession: PRJEB85302) were mapped to the jmh43 hankyphage prophage using bwa mem v. 0.7.17-r1188 [74], following delimitation of the prophage ends using capsid data. If the prophage had integrated into a different chromosomal region, reads at the ends would be clipped. Soft-clipped reads were extracted with the Pysam software package [75]. The clipped regions ($\geq$ 15 nt) were subsequently searched into the jmh43 bacterial chromosome using blastn (-task blastn-short -qcov_hsp_perc 100 -perc_identity 100). Clipped reads that were mapped into multiple genome locations were discarded (<1%). To quantify new junction locations, we considered only reads clipped at the start of the HP and recorded the last chromosomal position of each aligned read. The plot of integration spots across the jmh43 genome was generated with pyCirclize [76].

## Mutations at VR region

DGR components (RT, TR and VR regions) were detected as previously from genome assemblies [77]. Briefly, hmmsearch v. 3.4 was used to detect RT based on matches to HMM profiles of curated RT. Then, repeats around the candidate RT were detected with blastn (option -word_size 8 -dust no -gapopen 6 -gapextend 2). Finally, the selection of the TR and target genes were based on the repeat patterns (e.g., mutation bias and distance to the RT). The pipeline used is available at https://bitbucket.org/srouxjgi/dgr_scripts/src/master/DGR_identification/. The target region, which includes the VR, was aligned with mafft v.7 [78]. The resulting nucleotide alignment was used to design conserved primers flanking the VR region of the clinical isolates (F: "ATATCCACTGCCTCC", R: "GCCAGACGAGAACCG"). Primers were used to specifically collect reads spanning the complete VR region using SeqKit v.2.3 amplicon function [79]. The resulting amplicons were aligned with mafft, and the genotypes analyzed with R. To distinguish between DGR activity and background mutations/errors, we computed nucleotide diversity at each position as $\pi = 1 - (A^2 + T^2 + C^2 + G^2)$. We considered a VR to be diversified in a given strain if $\text{mean}(\pi,A) > \text{mean}(\pi,\text{nonA}) + 2 * \text{sd}(\pi,\text{nonA})$ [17]. Alternatively, we used a mapping approach to include also reads that span some region of the VR. Briefly, reads matching VR flanking primers were first filtered using bbduk.sh [80] (bbduk options $k = 15$ mm = f). The resulting reads were mapped against the VR region $\pm 100$ nt using bbmap to obtain a global alignment of the reads instead of the soft-clipped alignment of local alignments (bbmap options "vslow minid=0") [81]. A mpileup file was obtained using samtools mpileup (options -f and -A to annotate reference alleles and keep anomalous read pairs, respectively). A custom R script was then used to obtain allelic frequencies and nucleotide diversity from the raw mpileup file.

## Transmission Electron Microscopy (TEM)

All micrographs shown in the main text were prepared by mixing the lysates with paraformaldehyde (4% final concentration) for 24 hours and then incubating 4 μl of lysate for 1 min on CF300 carbon-coated grids (previously glow discharged 2 mA, 1 min in ELMO). The grids were rinsed four times in MilliQ water drops and contrasted using 2% uranyl acetate. Images were acquired with a Tecnai (ThermoFisher) T12 operating at 120 kV and a RIO 13 camera (Ametek).

For the micrographs in S5 Fig and S15 Fig, 4 μl of lysate was incubated for 1 min on CF300 carbon-coated grids (previously glow discharged 2 mA, 1 min in ELMO). The grid was rinsed four times in MilliQ water drops and contrasted using 2% uranyl acetate. Images were acquired with a Tecnai (ThermoFisher) T12 operating at 120 kv and a RIO 13 camera (Ametek). Particle size measurements were performed using FIJI [82].

## Analytical ultracentrifugation (AUC)

Overnight cultures were diluted to an absorbance of OD600 = 0.05 and grown anaerobically in 15 mL cell culture tubes at 37 °C until exponential phase. The cultures were then incubated anaerobically at 37 °C for 18 h with or without addition of IPTG (500μM). The cultures were then centrifuged at 7,142*g* for 15 min at 4 °C, the supernatant was then filtered using 0.45 μm filters. Lysates were then concentrated, and medium exchanged for PBS (no calcium nor magnesium) by placing the lysate in amikon 100 kDa cutoff columns and spinning for 5 m at 3,000*g* at 4 °C or until the sample was concentrated to 500 μl twice, keeping the top fraction. 130 μl of concentrated lysate was mixed with DNase buffer (1× final concentration) and treated with DNase Turbo (3 μl) for 2 h at 37 °C. The DNase treated lysates were then resuspended in 1 mL of PBS and spun down in new 15 mL Amikon 100 KDa columns to a final 500 μl volume to remove the DNase enzyme. Samples were centrifuged at 6,000 rpm in an Optima AUC analytical ultracentrifuge (Beckman

Coulter), at 20 °C in an eight-hole AN 50–Ti rotor equipped with 12-mm double-sector aluminum epoxy centerpieces. Detection of the biomolecule concentration as a function of radial position and time was performed by absorbance measurements at 260 nm, 280 nm and by interference detection. Ultracentrifugation experiments were performed in PBS. Sedimentation velocity data analysis was performed by continuous size distribution analysis c(s) with invariant D [83] using Sedfit 16.36 software. All the c(s) distributions were calculated with a fitted diffusion and a maximum entropy regularization procedure with a confidence level of 0.68. To evaluate the amount of DNA 260/280 absorbance ratio were calculated at the top of each peak; to note ratio below 1 shows the presence of less than 5% of DNA sample. Therefore, for ratio below 1 the amount of DNA is negligeable [84]. Buffer viscosity and density were calculated from the Sednterp software. Modeling of the different assemblies was evaluated with dimensions measured by TEM using spherical modeling as suggested by previous literature [85,86].

## Mass spectrometry

PEG-precipitated capsids (see '*capsid DNA extraction and sequencing*' section) from one biological replicate of a jmh43::*dcas9-repc*$_{HP}$ lysate were mixed with 1× Laemli buffer (Bio-Rad #1610747) and boiled at 100 °C for 10 min. After separation by denaturing SDS–PAGE using precast TGX 4%–15% gradient gels (Bio-Rad), the gel bands of interest were cut and washed using acetonitrile (ACN)/water-based dehydration/rehydration steps. Disulfide reduction was performed by adding 200 µL of a solution containing 10 mM dithiothreitol in ammonium bicarbonate (AMBIC) 50 mM to dehydrated gel bands for 30 min at 56 °C. Thiol alkylation was performed by incubating the previous samples in 20 µL of a 500 mM iodoacetamide solution in AMBIC for 30 min at room temperature. Reagents were removed using the previous washing procedure. Proteins were digested with a solution of 0.05 µg/µL of trypsin/Lys-C (Promega) in 100 µL of AMBIC overnight at 37 °C. The resulting peptide mixture was filtered and acidified with trifluoroacetic acid at a final concentration of 0.1% (v/v). For each sample, 6 µL were loaded on a C18 cartridge (Dionex Acclaim PepMap100, 5 µm, 300 µm i.d. × 5 mm) and eluted on a capillary reversed-phase column (Waters nanoEaze *M/Z* Peptide CSH C18, 1.8 µm, 75 µm i.d. × 25 cm) at 220 nL/min, with a gradient of 2% to 38% of buffer B in 60 min (buffer A: 0.1% aq. Formic Acid/ACN 98:2 (v/v); buffer B: 0.1% aq. Formic Acid/ACN 10:90 (v/v)), using a nanoRSLC U3000 system coupled with a Tribrid Orbitrap Eclipse mass spectrometer (ThermoFisher Scientific) using a "Top Speed" data-dependent acquisition MS experiment: 1 survey MS scan (400−2,000 *m/z*; resolution 70,000) followed by the highest possible number of MS/MS scans within 3 s, in the linear ion trap (dynamic exclusion of 30 s). Protein identification was performed with Proteome Discoverer 2.4 using SEQUEST against the jmh43 genome, with the following parameters: methionine oxidation, asparagine/glutamine deamidation as variable modifications, cysteine carbamidomethylation as fixed modification, MS1 error tolerance set at 10 ppm and MS/MS error tolerance at 0.6 Da, protein identification at FDR 1%. The identified proteins were compared against the proteome of the jmh43 strain using blastp (-e-value 1E−4).

## Capsid DNA extraction and sequencing

Lysates were mixed with PEG8000 1 M NaCL 10% (final concentration) to a final volume of 250 mL and let precipitate overnight at 4 °C. The lysates were then centrifuged at 20,000*g* for 2–3 h, and the pellet was resuspended by pipetting in 1 mL of DNase buffer and then placed in a shaker at 37 °C for 30 min to fully resuspend the pellet. Then 96 µl high titer purified phage were mixed with 2 µl of TURBO DNAse (Thermo Fisher, REF #AM2238) and 2 µl of benzonase (Abcam, REF #ab270049) and incubated at 37 °C for 1 h. The DNase was then inactivated by adding EDTA 15 mM (final concentration) and a heat-treatment for 10 min at 75 °C. Then, a proteinase

K treatment (3,7 mg/mL final concentration) enhanced with SDS (0.74% final concentration) at 55 °C for 45 min was performed to degrade phage capsids. For extraction of the remaining DNA, the PureGene DNA extraction kit (Qiagen) with the protocol for Gram-negative bacteria DNA extraction starting from step 8 (adding directly 100 µl of protein precipitation solution to our 100 µl sample) was used. An overnight DNA precipitation step was performed at 4 °C and the ethanol washes were performed on the next day. DNA was then resuspended for 1 h at 60 °C.

### Library preparation and sequencing

For the jmh43 capsids sample, libraries were generated manually following the manufacturer's protocol for Illumina TruSeq Nano DNA low-throughput library preparation kit (Illumina, San Diego, USA, REF #20015964) with a PCR step of 14 cycles. Briefly, samples were normalized to 500 ng DNA and sheared to 350 pb by sonication with a Covaris E220. AMPure XP beads were used for cleanup and size selection, and then adapters were ligated. Fragment sizes for all libraries were measured using a fragment Analyzer (Agilent) and pooled prior to sequencing. Finally, 1200 pM of the pooled libraries were sequenced on Nextseq2000 Illumina sequencer using a P1 flowcell. Sequencing was performed as pair-end 300 cycles.

For the jmh43::*dcas9-repc*$_{HP}$ capsids sample, the Novogene NGS DNA Library Prep Set (Cat No.PT004) was used for library preparation. Index codes were added to each sample. Briefly, the genomic DNA was randomly fragmented to size of 350bp. DNA fragments were end polished, A-tailed, ligated with adapters, size selected and further amplified by rolling circle amplification. PCR products were then purified (AMPure XP system), their size distribution was assessed by Agilent 2100 Bioanalyzer (Agilent Technologies, CA, USA), and they were quantified using real-time PCR. Libraries were sequenced on an Illumina Novaseq 6000 S4 flowcell with a PE150 strategy.

### Capsid DNA analysis

Raw reads from the viral faction (data available at ArrayExpress, accession: E-MTAB-14979) were mapped to the jmh43 chromosome and prophage genome using bwa mem v. 0.7.17-r1188. Samtools v.1.18 was used to obtain bam files and sequence depth [75]. Depth normalization was performed by dividing by the total number of mapped reads on that region.

### Capsid concentration by ultracentrifugation

Overnight cultures were diluted to an absorbance of $OD_{600} = 0.05$ and grown anaerobically in 5 mL cell culture tubes at 37 °C overnight. The next morning, the cultures were centrifuged at 7142*g* for 15 min at 4 °C, the supernatant was then filtered using 0.45 µm filters. 1.5 mL of each filtrate were pelleted by ultracentrifugation in Beckman coulter 1.5 mL tubes (REF #357448) using an Optima max XP ultracentrifuge with a fixed angle rotor (TLA-55) for 30 min at 50,000 RPM. The pellets were then resuspended in phage buffer (150 mM NaCl, 40 mM Tris-Cl, pH 7.4, 10 mM $MgSO_4$) and subsequently DNAse treated and heat-treated and used for the ddPCR.

### Phylogenetic analyses of *Bacteroides* and hankyphage-like prophages

Both datasets of 26 hankyphage genomes and 161 *Bacteroides* genomes were separately processed using the PanACoTA pipeline, version 1.4.0 [31]. The "prepare" step was executed with parameters –min 1e06 (to discard very similar genomes, based on mash distance), --max 0.8 (to discard very different genomes, based on mash distance) and –l90 380 (to discard genomes with high L90 values); the "annotate" step was executed with the parameter –l90 380 (same as before); the "pangenome" step was executed with the parameter -i 0.8 (percentage identity to cluster two proteins in the same family is 80%); the "persistent" step was executed with the parameter -t 0.9

(to select persistent genes found in at least 90% of the genomes); and the "tree" step was executed with the parameter -b 1,000 (to compute the tree using 1,000 bootstraps). The remaining parameters across the different steps of the pipeline were left as default. After the filtering steps of PanACoTA, all hankyphage genomes were kept, and 159 out of the initial 165 Bacteroides genomes were kept. The respective phylogenetic trees were rooted at their midpoint, displayed using iTOL [87] and posteriorly edited to include the tanglegram. Tree files for *Bacteroides* and hankyphages and their respective IDs can be found in supplementary S17 Data, S18 Data and S4 Data respectively, also available at https://figshare.com/s/3ff18cc2f6cc1edab0ae.

## Analysis of metaviromes

Metaviromes with more than 10,000 reads mapped to the reference hankyphage p00 [10] were downloaded and assembled with SPAdes v. v3.15.5 (--metaviral) [88]. Metaviromes containing the hankyphage in less than three contigs were selected for further analysis. Reads from each virome were then mapped to its correponding hankyphage using bwa mem v. 0.7.17-r1188 and the sequence coverage estimated with samtools depth [75].

## Supporting information

**S1 Fig. Characterization of the laboratory collection of *Bacteroides thetaiotaomicron* hankyphages.** (**a**) Confirmation of hankyphage presence by PCR using primers located on the hankyphage reverse transcriptase (S2 Table, primers 53,54). Plus and minus signs indicate positive (+) and negative (−) controls for hankyphage DNA. (**b**) Alignment of the jmh63 genome on the hankyphage p00 genome using the Geneious software, suggesting that the jmh63 hankyphage coud be fragmented.
(TIF)

**S2 Fig. Validation of virion quantification by ddPCR.** (**a**) Effect of DNase treatment of phage lysates using the two ddPCR primer pairs. The *rplB* gene is not detected after DNase treatment of jmh43 lysates. (**b**) Representative ddPCR result from a jmh43 DNase treated lysate. Error bars represent the uncertainty values calculated by the Naica reader. The reaction is optimized to quantify hankyphage DNA (HP) copies and has low uncertainty values. The quantification of *rplB* falls below the minimum concentration threshold to detect two separate clouds and has high uncertainty values, indicating low or no genomic DNA presence. The individual quantitative values underlying this figure can be found in the S5 Data file also available at: https://figshare.com/s/3ff18cc2f6cc1edab0ae.
(TIF)

**S3 Fig. Transcriptional effects of *repc*$_{HP}$ silencing in *Bacteroides thetaiotaomicron* jmh43.** (**a**) Representative coverage plots of one of the three transcriptomic biological replicates at different time points of both jmh43::*dcas9-repc*$_{HP}$ and jmh43::*dcas9-nt*. The *x*-axis corresponds to the hankyphage genome, genes encoded in the positive orientation are colored in purple and those in the antisense orientation are coloured in blue. Coverage plots were produced using the Integrated Genome Browser (IGB) software. (**b**) Normalized transcriptomic coverage of the hankyphage genome of jmh43::*dcas9-repc*$_{HP}$ (red) and jmh43::*dcas9-nt* (blue) strains at different timepoints after IPTG-induced RepC$_{HP}$ silencing. Stars indicate significant differential up-regulation (green) or downregulation (red) of the gene in jmh43::*dcas9-repc*$_{HP}$ compared to jmh43::*dcas9-nt*. Coverage is normalized by total reads per sample and gene length. Values of 0 were converted to 1 for their visualization in the log$_{10}$ scale. Differential gene expression analysis was performed using Deseq2. (**c**) Log2 fold change in dCas9 expression between jmh43::*dcas9-repc*$_{HP}$ and jmh43::*dcas9-nt* control. This indicates that the jmh43::*dcas9-nt*

control expresses dCas9 less than jmh43::*dcas9-repc*HP. (**d**) Superposition of an Alphafold2 model of RepC~HP~ (yellow) (modeled in dimer, only one monomer shown) and the crystal structure of the λ repressor CI (blue) (dimer, only a monomer shown). RMSD: root mean square of atomic positions. The individual quantitative values underlying this figure can be found in the S6 Data file also available at: https://figshare.com/s/3ff18cc2f6cc1edab0ae. (PNG)

**S4 Fig. DGR activity in jmh43 and jmh43::*dcas9-repc*HP virions** . (**a**) Bar plots indicate the proportion of A, C, T, and G nucleotides at the VR (highlighted in blue) and flanking region which differ from the reference VR after using a global mapping approach (see methods). Of note, jmh43::*dcas9-repc*HP lysate was sequenced using a PCR-free kit, whereas for jmh43 a previous step of PCR amplification was required to obtain enough DNA for Illumina sequencing. As such, background mutation/noise, considerably differ between both samples. Positions targeted by the DGR system are highlighted in dark blue. (**b**) Summary statistics of non-reference allele data (mut) presented in panel a depending on the region considered. A in TR denotes positions expected to be diversified by the DGR compared to non-A positions. As a control, diversity around the VR flanking region (±100 nt) is also shown. Mean_mut and sd_mut denotes the average and standard deviation of non-reference allele, respectively, and coverage indicates the average mapping depth used for estimating alternative allele frequencies. Significant DGR activity was only found for *jmh43::dcas9-repc*HP (mean diversity in targeted positions> mean diversity in non-targeted positions + 2 sd). The individual quantitative values underlying this figure can be found in the S7 Data file also available at: https://figshare.com/s/3ff18cc2f6cc1edab0ae. (TIFF)

**S5 Fig. TEM microscopy image of the lysate of *P. dorei HM719*.** Negative staining and TEM microscopy of the reference *P. dorei HM719* lysate revealed very few particles and absence of tails. (TIF)

**S6 Fig. Capsid sizes.** Box and violin plot showing the particle diameter from the TEM micrographs of the jmh43 and jmh43::*dcas9-repc*HP lysates, including tailed and non-tailed particles. Individual values shown as points, boxes indicate the median and bars indicate the minimum and maximum values. Significant differences in sizes were calculated via a non-parametric Kruskal–Wallis test and are shown as asterisks (the two asterisks correspond to a *p*-value between 0.05 and 0.005). The individual quantitative values underlying this figure can be found in the S8 Data file also available at: https://figshare.com/s/3ff18cc2f6cc1edab0ae. (TIF)

**S7 Fig. Expression of *Bacteroides thetaiotaomicron* jmh43 putative prophages.** Normalized transcriptomic coverage of the hankyphage genome and other putative prophages of jmh43::*dcas9-repc*HP at different timepoints after IPTG-induced *repc*HP silencing. Coverage is normalized by total reads per sample and gene length. Values of 0 were converted to 1 for their visualization with a $Log_{10}$ scale. The individual quantitative values underlying this figure can be found in the S9 Data file also available at: https://figshare.com/s/3ff18cc2f6cc1edab0ae. (TIFF)

**S8 Fig. Supporting results on hankyphage limited lysis and absence of transduction.** (**a**) Growth curves of *Bacteroides thetaiotaomicron* VPI5482 capsule mutants exposed to lysates of jmh43ΔHP or jmh43::*dcas9-repc*HP, Curves are the average of two biological replicates. The individual quantitative values underlying this figure can be found in the S10 Data file. (**b**)

Representative spot assay photographs, overlays of different capsule mutant strains with spots of jmh43::*dcas9-repc*$_{HP}$ (RepcHP) or jmh43ΔHP lysates, no lysis plaques could be observed. (**c**) Illustration representing the location of insertion of *ermG* (orange) within the hankyphage genome within the jmh43 strain. (**d**) PCR checks indicating the absence of *ermG* acquisition of different colonies (1–8) as well as a positive (+) and negative (−) control. (PNG)

**S9 Fig. Analysis of the hankyphage DNA ends.** (**a**) Sequence coverage of DNA extracted from the hankyphage capsids after mapping to the jmh43 chromosome region where the phage is initially located. Red dashed lines mark the beginning (5′) and end (3′) of the phage. (**b**) Zoom in on the 5′ end of the phage. (**c**) Distribution of the size of chromosomal fragments packaged at the 5′ end of the phage. (**d**) Zoom in on the 3′ end of the phage shows decreasing coverage of jmh43 chromosome over a ~ 2,300 nucleotide region indication packaging of a variable size fragment of the chromosome on the 3′ end of the phage of approximately 1–2.3kb. The individual quantitative values underlying this figure can be found in the S11 Data file also available at: https://figshare.com/s/3ff18cc2f6cc1edab0ae. (TIF)

**S10 Fig. Generalized uneven hankyphage genome coverage in capsids of HP+ strains.** Bar plot showing the ratio of hankyphage genome right- and left-end copies quantified by ddPCR on PEG-precipitated capsids of HP+ strains. Error bars depict the standard deviation from the mean of two or three biological replicates (individual values plotted as points). The individual quantitative values underlying this figure can be found in the S12 Data file also available at: https://figshare.com/s/3ff18cc2f6cc1edab0ae. (TIFF)

**S11 Fig. Limited hankyphage burden on HP+ strains.** Growth curves performed on three HP+ strains and their respective mutant deleted for the hankyphage (three biological replicates). The individual quantitative values underlying this figure can be found in the S13 Data file also available at: https://figshare.com/s/3ff18cc2f6cc1edab0ae. (TIF)

**S12 Fig. Supporting data for the phylogenetic trees.** (**a**) wGG and (**b**) Sorensen-Dice matrices supporting the hankyphage phylogenetic tree from Fig Fig 4. (TIF)

**S13 Fig. Comparison of the jmh43 hankyphage coverage pattern with hankyphages assembled from metaviromes.** Coverage plots showing the patterns of hankyphage DNA coverage in four representative viral metagenomic samples as well as in jmh43 laboratory-extracted lysates. The gradual increase of coverage from the 5′ to the 3′ in the metaviromes may be attributed to the use of sequencing kits that use transposases (Nextera) to ligate the adapters in this study [89]. In the case of hankyphages, this might prevent the ligation of the adaptors at the phage 5′ but not at the 3′ end given the differential amount of host DNA that is packaged at each side. Coverage is normalized by the sum of the mapped reads to that region. The individual quantitative values underlying this figure can be found in the S14 Data file also available at: https://figshare.com/s/3ff18cc2f6cc1edab0ae. (TIFF)

**S14 Fig. Expression of putative anti-phage defense systems in *Bacteroides* thetaiotaomicron jmh43.** (**a**) Normalized transcriptomic coverage of the putative anti-phage defense systems of jmh43::*dcas9-repc*$_{HP}$ at different timepoints after IPTG-induced *repc*$_{HP}$ silencing. Coverage is normalized by total reads per sample and gene length. (**b**) Normalized

transcriptomic coverage of the putative anti-phage defense systems of jmh43::*dcas9-repc*$_{HP}$ and jmh43::*dcas9-nt* at 22 hpi indicating the significant differences in expression calculated by the differential expression analysis. This was the only time point where significant differences in expression were detected for these genomic regions. The individual quantitative values underlying this figure can be found in the S15 Data file also available at: https://figshare.com/s/3ff18cc2f6cc1edab0ae.
(TIFF)

**S15 Fig. TEM microscopy image of the lysate of jmh43Δbil strain.** Negative staining and TEM microscopy of a mutant of the jmh43 strain where the putative bil defense system was deleted (jmh43Δbil).
(TIF)

**S1 Table. Laboratory collection of Bacteroides isolates gathered from French hospitals.** List of *Bacteroides* laboratory collection indicating whether they contain the hankyphage (HP+) (green) or not (HP−) (red) and whether they were conjugable (+) or not (−), as well as their resistance or sensitivity to erythromycin (ermR, ermS) or tetracycline (TetR, TetS). Also available at https://figshare.com/s/e704aa15d732cbf7d1a4.
(XLSX)

**S2 Table. Primers and probes list.** Also available at https://figshare.com/s/e704aa15d732cbf7d1a4.
(XLSX)

**S3 Table. *Bacteroides thetaiotaomicron* hankyphages from the laboratory collection.** Table indicating the hankyphages from the laboratory collection as well as their taxonomy (extracted using VIRIDIC), TR and VR sequences and their nucleotide identity to the p00 phage. Also available at https://figshare.com/s/e704aa15d732cbf7d1a4.
(XLSX)

**S4 Table. Mutant strains used in this study.** Also available at https://figshare.com/s/e704aa15d732cbf7d1a4.
(XLSX)

**S5 Table. Predicted prophages in jmh43 genome.** Also available at https://figshare.com/s/e704aa15d732cbf7d1a4.
(XLSX)

**S6 Table. Mass Spectrometry data against hankyphage proteins.** Table showing the hankyphage proteins present in PEG-precipitated jmh43::*dcas9-repc*$_{HP}$ capsids. The proteins are ordered by gene position in the hankyphage genome starting from the 5′ end. The four columns on the right indicate in which of the four different SDS-PAGE band fractions the proteins were detected at (ranging from >250 kDa–10 kDa). Presence detected with high confidence is indicated by 'high'. Also available at https://figshare.com/s/e704aa15d732cbf7d1a4.
(XLSX)

**S7 Table. Putative anti-phage defense systems in jmh43.** Table indicating the putative defense systems in the jmh43 strain detected using Defense-Finder. The table also indicates the number of putative genes present belonging to the system (gene count column) and the names of the genes (genes in the system column). Also available at https://figshare.com/s/e704aa15d732cbf7d1a4.
(XLSX)

**S1 Data. Raw values for** Fig 1. (**A**) Supplementary dataset for Fig 1b—GR diversification in Bacteroides hankyphages. (**B**) Raw data for Fig 1c—DGR diversification in Bacteroides hankyphages. (**C**) Raw data for Fig 1d—ddPCR bacteroides strains. (**D**) Raw data for Fig 1e—ddPCR inductions.
(XLSX)

**S2 Data. Raw values for** Fig 2. (**A**) Raw data for Fig 2a—Log2-fold change in gene expression of hankyphage genes in jmh43::dcas9-*repc*_HP strain compared to the jmh43::*dcas9-nt* strain. (**B**) Raw data for Fig 2c—Transposition events. (**C**) Raw data for Fig 2e—CFU. (**D**) Raw data for Fig 2e—CFU.
(XLSX)

**S3 Data. Raw values for** Fig 3. (**A**) Raw data for Fig 3b—Analytical ultracentrifugation. (**B**) Raw data for Fig 3d—Capsid DNA coverage. (**C**) Raw data for Fig 3e—Genomic DNA coverage. (**D**) Raw data for Fig 3f—ddPCR ultracentrifuged capsids. (**B**) Raw data for Fig 2c—Transposition events. (**C**) Raw data for Fig 2e—CFU. (**D**) Raw data for Fig 2e—CFU.
(XLSX)

**S4 Data. Supporting data for** Fig 4. (**A**) NCBI genomes used for the phylogenetic trees. (**B**) IDs of *Bacteroides* host used for the phylogenetic trees. (**C**) IDs of hankyphages used for the phylogenetic trees.
(XLSX)

**S5 Data. Raw values for S2B Fig.**
(XLSX)

**S6 Data. Raw values for S3 Fig.** (**A**) Raw values for S3B Fig—RNAseq hankyphage. (**B**) Raw values for S3C Fig—Log2 Fold change in dCas9 expression.
(XLSX)

**S7 Data. Raw values for S4 Fig.**
(XLSX)

**S8 Data. Raw values for S6 Fig.**
(XLSX)

**S9 Data. Raw values for S3B Fig.**
(XLSX)

**S10 Data. Raw values for S8A Fig.**
(XLSX)

**S11 Data. Raw values for S9 Fig.** (**A**) Raw values for S9a Fig. (**B**) Raw values for S9b Fig. (**C**) Raw values for S9c Fig. (**D**) Raw values for S9d Fig.
(XLSX)

**S12 Data. Raw values for S10 Fig—ddPCR PEG-precipitated capsids.**
(XLSX)

**S13 Data. Raw values for S11 Fig—Growth curves.**
(XLSX)

**S14 Data. Raw values for S11 Fig—Hankyphage coverage in metagenomic data.**
(XLSX)

**S15 Data. Raw values for S14 Fig.** Expression of putative defence systems in the *jmh43::dcas9-repc$_{HP}$* strain and the *jmh43::dcas9-nt* strain.
(XLSX)

**S16 Data. Differentially expressed genes in the jmh43::dcas9-*repc*$_{HP}$ strain versus the jmh43::*dcas9-nt* strain at different timepoints.**
(XLSX)

**S17 Data. *Bacteroides* phylogenetic tree text file.**
(TXT)

**S18 Data. Hankyphage phylogenetic tree text file.**
(TXT)

## Acknowledgments

We thank Christophe Beloin, Ariane Toussaint, Pascale Boulanger, Marie-Agnès Petit and Mart Krupovic for critical reading of the manuscript and Raphael Laurenceau for insightful discussions. We thank Azimdine Habib and Laurence MA for the RNA and DNA sequencing performed at the Institut Pasteur Biomics Platform. We further thank Laure Lemée for her contribution to the RNAseq analysis as well as Gérard Pehau Arnaudet for the TEM performed at the Microscopy Platform at the Institut Pasteur. We would also like to thank Giovanni Chiappetta for processing the MS samples.

## Author contributions

**Conceptualization:** Sol Vendrell-Fernández, David Bikard, Jean-Marc Ghigo.

**Data curation:** Sol Vendrell-Fernández, Beatriz Beamud.

**Formal analysis:** Sol Vendrell-Fernández, Beatriz Beamud, Yasmina Abou Haydar, Jorge Am de Sousa, David Bikard.

**Funding acquisition:** David Bikard, Jean-Marc Ghigo.

**Investigation:** Sol Vendrell-Fernández, Beatriz Beamud, Yasmina Abou Haydar.

**Methodology:** Sol Vendrell-Fernández, Beatriz Beamud, Julien Burlaud-Gaillard, Etienne Kornobis, Bertrand Raynal, Joelle Vinh, David Bikard, Jean-Marc Ghigo.

**Project administration:** Jean-Marc Ghigo.

**Supervision:** David Bikard, Jean-Marc Ghigo.

**Validation:** Sol Vendrell-Fernández, David Bikard, Jean-Marc Ghigo.

**Visualization:** Sol Vendrell-Fernández, Beatriz Beamud, Jorge Am de Sousa, Julien Burlaud-Gaillard, Etienne Kornobis, Bertrand Raynal.

**Writing – original draft:** Sol Vendrell-Fernández, Jean-Marc Ghigo.

**Writing – review & editing:** Sol Vendrell-Fernández, Beatriz Beamud, Yasmina Abou Haydar, Jorge Am de Sousa, Joelle Vinh, David Bikard, Jean-Marc Ghigo.

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
