## [Editor Report · Decision Letter 0]

1 Aug 2024

Dear Dr Bikard, 

Thank you for submitting your manuscript entitled "Incomplete lytic cycle of a widespread Bacteroides bacteriophage leads to the formation of defective viral particles" for consideration as a Research Article by PLOS Biology. 

Your manuscript has now been evaluated by the PLOS Biology editorial staff, as well as by an academic editor with relevant expertise, and I am writing to let you know that we would like to send your submission out for external peer review as a *Short Report*.

However, before we can send your manuscript to reviewers, we need you to complete your submission by providing the metadata that is required for full assessment. To this end, please login to Editorial Manager where you will find the paper in the 'Submissions Needing Revisions' folder on your homepage. Please click 'Revise Submission' from the Action Links and complete all additional questions in the submission questionnaire. Please, when adding the rest of the metadata choose "Short Report".

Once your full submission is complete, your paper will undergo a series of checks in preparation for peer review. After your manuscript has passed the checks it will be sent out for review. To provide the metadata for your submission, please Login to Editorial Manager (https://www.editorialmanager.com/pbiology) within two working days, i.e. by Aug 03 2024 11:59PM.

Kind regards,

Melissa

Melissa Vazquez Hernandez, Ph.D.

Associate Editor

PLOS Biology

---

## [Decision Letter · Decision Letter 1]

11 Sep 2024

Dear Dr Bikard,

Thank you for your patience while your manuscript "Incomplete lytic cycle of a widespread Bacteroides bacteriophage leads to the formation of defective viral particles" was peer-reviewed at PLOS Biology. It has now been evaluated by the PLOS Biology editors, an Academic Editor with relevant expertise, and by several independent reviewers. 

In light of the reviews, which you will find at the end of this email, we would like to invite you to revise the work to thoroughly address the reviewers' reports. The reviewers are positive about the relevance of the findings but have raised several concerns that need to be addressed before we can consider the study further. Reviewer #1 suggests reorganizing the manuscript to avoid potential misinterpretations and recommends testing alternative methods to concentrate or purify the virions. Reviewer #2 suggests evaluating whether host genes are differentially expressed in the HP knockout strain and determining if other prophages are transcribed in its absence. S/he also highlights the lack of replicates in figure 4b and questions the claim that HP generally cannot infect in lab settings. The reviewer advises either lowering the strength of this claim or providing evidence from at least two additional phages from the collection. 

IMPORTANT: while we think the mouse experiments could add valuable information, this are not required for publication. All other concerns and suggestions from the reviewers, should be addressed.

Given the extent of revision needed, we cannot make a decision about publication until we have seen the revised manuscript and your response to the reviewers' comments. Your revised manuscript is likely to be sent for further evaluation by all or a subset of the reviewers.

**IMPORTANT - SUBMITTING YOUR REVISION**

*Re-submission Checklist*

*Published Peer Review*

*PLOS Data Policy*

*Blot and Gel Data Policy*

Sincerely,

Melissa

Melissa Vazquez Hernandez, Ph.D.

Associate Editor

PLOS Biology

REVIEWERS' COMMENTS:

Reviewer #1: Andrey Shkoporov

Reviewer #1: 

This manuscript describes investigation into certain aspects of lifestyle of Hankyphage - a temperate virus infecting Bacteroides, one of the most abundant and prevalent bacteria in the human gut. Since current knowledge on Bacteroides phages is limited, and the phage in question employing a somewhat unusual replicative transposition mechanism for its replication, the results presented in this manuscript present considerable interest to the human microbiome research community, as well as to phage biologists.

Manuscript structure:

Results presented under the sub-section "Abnormal hankyphage virion assembly is partly recovered by repCHP silencing" invalidate preliminary conclusions that the reader might arrive to while reading the previous sub-section "Production of hankyphage virions does not lead to detectable lysis nor transduction". Therefore I suggest reversing the order in which results are presented and merging both subsections into one.

Specific comments:

line 126: "Seven B. thetaiotaomicron strains out of the 22 (32%) hankyphage-like prophages, hereafter called HP+ strains..." - this is unclear, please revise

line 387: "The law abundance..." - low abundance?

line 415: to me this is more indicative of virion damage during PEG precipitation, partial ejection of DNA, and its degradation during the subsequent DNase treatment. Please consider this as an alternative explanation. Also trying alternative methods to concentrate/purify virions could be a great addition to this work.

line 986: ref 10 looks corrupted

Reviewer #2: 

Vendrell-Fernández et. al describe the characterization of a lysogenic Hankyphage in Bacteroides thetaiotaomicron. Using of variety of genetic, genomic, and biochemical techniques, they demonstrates the lysogen is spontaneously induced in regular laboratory culture. However, the resultant phage is seemingly defective, potentially due to incorrect virion assembly or genomic packaging. Diversity generating retroelements as well as a master repressor were identified in the phage and bioinformatic analysis revealed its prevalence in Bacteroides strains despite its inactivity in vitro. Altogether, the manuscript is a compelling set of negative data that points to the mysterious nature of Bacteroides phages that often do not follow the same set of rules as their better studied Proteobacteria and Firmicutes counterparts. 

Major Comments

Line 363 to 407 - The evidence that other prophages within the B. theta strain are not affecting Hankyphage production and maturation could be strengthen. The lack of DNA packaging in incomplete virions as well as the lack of proteins via mass spectroscopy does not show that the other prophage are not contributing to the incomplete pieces seen via TEM. Showing that transcription of these other prophage genes genes is not occurring would be stronger evidence that these components are only from the hankyphage. By the same token, it would be interesting to assess in the HP knockout strain if host genes are differentially expressed or if another prophage engages in transcription in the absence of HP. Even in the reported RNAseq dataset in Figure 3B, it would be important to note if knockdown of the repC repressor alters transcription of any genes outside of HP. For example, phage structural protein production could trigger a phage defense system that interferes with virion assembly.

Figure 4b - Only having one replicate is insufficient to make a fair comparison between the nt and repC dCas9 strains. One would expect transposition to be a stochastic event and additional replicates would be required to show that the extent of transcription is truly greater in the repC knockdown and not due to a jackpot effect in the nt control. I would suggest repeating this experiment with several more replicates to show this is a consistent occurrence.

Throughout - It is claimed that hankyphages as a whole are incapable of infecting and transducing in laboratory settings, however you only show data for a single phage for the vast majority of the paper (jmh43). These claims can only be applied to jmh43 and consistently applying them to hankyphages across B. theta or Bacteroides lacks substantial evidence. I suggest softening generalized claims about Hankyphage to focus on jmh43 or performing the experiments to include 2+ more phage from the collection of 7 that was obtained.

While likely beyond the scope of the manuscript, it would be interesting to see if: 1) Transduction could be observed in the mouse gut; 2) Transduction could be observed in ex vivo conditions, such as cecal extracts; 3) If different media types would yield viable virions, especially those that better resemble the ionic strength and pH of the gut environment where (presumably) these phages are active.

Minor Comments

Line 242 - Different citation format than the rest of the manuscript.

Line 449 - grammatical issue. Should be "(Sig 11 Fig), suggesting)"

Figure 1b - State total number of cultures used to determine nucleotide diversity 

Figure 4c - Why is there a dip in virions at 2 hours for one of the jmh43::dcas9-repCHP replicates?

---

## [Decision Letter · Decision Letter 2]

17 Jan 2025

Dear Dr Bikard,

Thank you for your patience while we considered your revised manuscript "Incomplete lytic cycle of a widespread Bacteroides bacteriophage leads to the formation of defective viral particles" for publication as a Short Reports at PLOS Biology. This revised version of your manuscript has been evaluated by the PLOS Biology editors, the Academic Editor and the original reviewers.

Based on the reviews and on our Academic Editor's assessment of your revision, we are likely to accept this manuscript for publication, provided you satisfactorily address the remaining editorial points. Please also make sure to address the following data and other policy-related requests.

a) The article type is a Short Reports. This type of article has a limit of 4 Figures, while you currently have 7 main figures. Please reduce the number of figures either by placing some together, or sending some to the Supplementary material.

Please supply the numerical values either in the a supplementary file or as a permanent DOI’d deposition for the following figures:

Figure 1B, 2AB, 4B, 5B (axis is mispelled from ciefficient→ coefficient), 6ABC, S2B, S3BC, S4B, S6, S7, S8A, S9AB, S10, S11, S13, S14AB

c) Please cite the location of the data clearly in all relevant main and supplementary Figure legends, e.g. “The data underlying this Figure can be found in S1 Data” or “The data underlying this Figure can be found in https://doi.org/10.5281/zenodo.XXXXX” 

d) Please provide the tree files for the phylogenetic trees in Figure 7

e) Please ensure that your Data Statement in the submission system accurately describes where your data can be found and is in final format, as it will be published as written there.

f) Per journal policy, if you have generated any custom code during the course of this investigation, please make it available without restrictions upon publication. Please ensure that the code is sufficiently well documented and reusable, and that your Data Statement in the Editorial Manager submission system accurately describes where your code can be found.

We expect to receive your revised manuscript within two weeks. 

*Published Peer Review History*

*Press*

Sincerely,

Melissa

Melissa Vazquez Hernandez, Ph.D.

Associate Editor

PLOS Biology

REVIEWERS' COMMENTS

Reviewer #2: The revisions addressed all of my previous concerns and I believe the manuscript is well suited for publication.

---

## [Editor Report · Decision Letter 3]

27 Jan 2025

Dear Dr Bikard,

Thank you for the submission of your revised Short Reports "Incomplete lytic cycle of a widespread Bacteroides bacteriophage leads to the formation of defective viral particles" for publication in PLOS Biology. On behalf of my colleagues and the Academic Editor, Jeremy J Barr, I am pleased to say that we can in principle accept your manuscript for publication, provided you address any remaining formatting and reporting issues. These will be detailed in an email you should receive within 2-3 business days from our colleagues in the journal operations team; no action is required from you until then. Please note that we will not be able to formally accept your manuscript and schedule it for publication until you have completed any requested changes.

IMPORTANT: Thank you for uploading all Supplementary Tables and the Supporting Data to Figshare. I would also like to thank you for adjusting the Figure legends to mention the location of the data. However, I would like to ask you for two things. 1) Please add in the Figure legends the link to the Figshare files. Currently, the link is not in an obvious location in the manuscript. You could say something like "The individual quantitative values underlying this figure can be found in the S1 data file in https://figshare.com/s/3ff18cc2f6cc1edab0ae". 2) In your Data Statement, please also provide both Figshare links for the Supplementary Tables and for the Supporting Data. I have asked my colleagues to include this request alongside their own.

PRESS

Sincerely,

Melissa 

Melissa Vazquez Hernandez, Ph.D., Ph.D.

Associate Editor

PLOS Biology
